# Optimizing Large Language Model Training Using FP4 Quantization

**Ruizhe Wang** [1 2 †]  **Yeyun Gong** [3 2]  **Xiao Liu** [3 2]  **Guoshuai Zhao** [3 2]  **Ziyue Yang** [3 2]
**Baining Guo** [3]  **Zhengjun Zha** [1]  **Peng Cheng** [3 2]

## Abstract

The growing computational demands of training large language models (LLMs) necessitate more efficient methods. Quantized training presents a promising solution by enabling low-bit arithmetic operations to reduce these costs. While FP8 precision has demonstrated feasibility, leveraging FP4 remains a challenge due to significant quantization errors and limited representational capacity. This work introduces the first FP4 training framework for LLMs, addressing these challenges with two key innovations: a differentiable quantization estimator for precise weight updates and an outlier clamping and compensation strategy to prevent activation collapse. To ensure stability, the framework integrates a mixed-precision training scheme and vector-wise quantization. Experimental results demonstrate that our FP4 framework achieves accuracy comparable to BF16 and FP8, with minimal degradation, scaling effectively to 13B-parameter LLMs trained on up to 100B tokens. With the emergence of next-generation hardware supporting FP4, our framework sets a foundation for efficient ultra-low precision training.

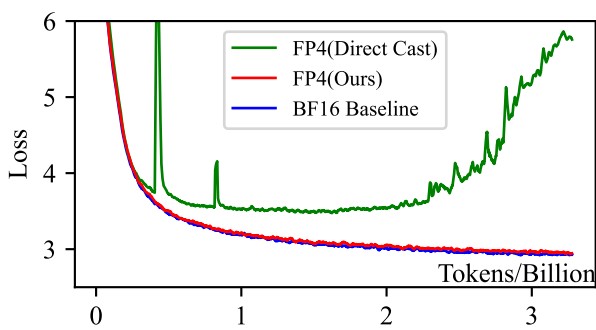

*Figure 1.* Directly casting to FP4 results in significantly higher training loss, whereas our proposed FP4 method achieves accuracy comparable to the BF16 baseline. These results are based on experiments with a 400M LLaMA2 model.

## 1. Introduction

In the past two years, the rapid development of large language models (LLMs) has significantly reshaped both research priorities and industrial practices. Theoretical analyses and empirical evidence consistently demonstrate that scaling up model size leads to substantial performance improvements (Kaplan et al., 2020; Bi et al., 2024). However, training such large-scale models poses considerable chal-

lenges, demanding extensive time, energy, and financial resources. For example, Llama 3 (Dubey et al., 2024) 405B is trained on up to 16K H100 GPUs for 54 days. Similarly, GPT-4 (Achiam et al., 2023), with an estimated 1T parameters, required an extraordinary amount of computational power. These examples highlight the urgent need for more efficient training methods to keep up with the increasing demands of LLM development.

Model quantization has proven to be an effective technique for reducing training costs, as low-bit arithmetic kernels can save memory and accelerate computations when used appropriately. Most LLM training systems traditionally rely on FP32 (full precision) or FP16/BF16 (half precision) data formats, but quantization enables these formats to be reduced to lower precision, such as 8-bit or even 4-bit.

Recent advancements in computational hardware, such as NVIDIA's H100 GPUs (Nvidia, 2023) and the upcoming B200 GPUs (Nvidia, 2024), have introduced support for low-bit arithmetic kernels, enabling more efficient computation. The Hopper series GPUs feature high-performance FP8 tensor cores, delivering a 2x speed-up compared to FP16 tensor cores. Meanwhile, the Blackwell series GPUs extend this capability by supporting FP6 and FP4 formats, with FP4 offering the potential to double computational throughput over FP8. Studies like FP8-LM (Peng et al.,

[†]Work done during internship in MSRA [1]University of Science and Technology of China [2]Microsoft SIGMA Team [3]Microsoft Research Asia. Correspondence to: Yeyun Gong <yegong@microsoft.com>, Peng Cheng <pengc@microsoft.com>.

2023) and NVIDIA's Transformer Engine (Nvidia, 2022) have demonstrated the feasibility of FP8 tensor cores for model training. But the application of FP4 tensor cores in model training remains an open research question.

However, leveraging 4-bit data formats for neural network training presents significant challenges due to the extremely limited bit width. Directly quantizing LLMs to such a low-bit format often results in substantial accuracy degradation, as shown in Figure 1. This is primarily because low-bit formats are constrained by a limited dynamic range, which increases the risk of overflow and underflow. Even existing methods for 8-bit quantization experience some degree of accuracy loss, underscoring the difficulties of employing a 4-bit format, which provides only 16 distinct representable values.

In this study, we pioneeringly propose a framework for training language models using the FP4 format, providing a validation of the feasibility of this ultra-low precision representation. To tackle the significant quantization errors associated with weights and activations during model training, we present a series of optimization techniques: (1) For weights, we present a differentiable quantization estimator to improve gradient updates in FP4 computations. By analyzing the impact of quantization on neural network forward and backward passes, we derive a function with correction terms for accurate gradient estimation; (2) For activations, we develop an outlier clamping and compensation strategy to address the issue of outlier values commonly observed during LLM training. By analyzing activation distributions in LLMs, we introduce a clamping method and a sparse auxiliary matrix to preserve quantization accuracy and maintain model performance.

We conduct comprehensive experiments to demonstrate that our FP4 training framework achieves accuracy comparable to models trained in BF16 or FP8 formats with the same hyperparameters. Leveraging the FP8 tensor cores of NVIDIA H100 GPUs to emulate FP4 computations, we train LLMs with up to 13B parameters and 100B training tokens, with minor training loss gap. For zero-shot evaluation on downstream tasks, model trained with FP4 show competitive results against BF16 models. We anticipate better speed performance gains with the availability of next-generation hardware like NVIDIA's B-series GPUs. Our training framework can be found at aka.ms/MS.AMP.

## 2. Preliminaries

According to the IEEE 754 standard (Kahan, 1996), a binary floating-point number consists of three components: a 1-bit sign (S), exponent bits (E), and mantissa bits (M). This is commonly represented as ExMy, where x and y denote the number of bits for the exponent and mantissa, respectively.

For example, FP16 uses E5M10 and BF16 uses E8M7. FP8 typically has two variants: E4M3 and E5M2. In our work, we adopt the E2M1 format for 4-bit floating-point representation, as defined in prior studies (Rouhani et al., 2023b;a), with 2 bits for the exponent and 1 bit for the mantissa.

Unlike integer (INT) quantization, floating-point (FP) quantization features uneven quantization intervals and a larger dynamic range. To quantize a high-precision tensor like FP16 to FP4, we employ the commonly used **absmax** method (Dettmers et al., 2022; Peng et al., 2023):

$$x_{\text{fp4}} = Q(x_{\text{fp16}} \cdot \gamma), \quad \gamma = \frac{\text{MAX}_{\text{fp4}}}{\max(|x_{\text{fp16}}|)} \quad (1)$$

Here, $\text{MAX}_{\text{fp4}}$ represents the maximum absolute value in the FP4 format, and $\gamma$ serves as the scaling factor. For the E2M1 configuration, $\text{MAX}_{\text{fp4}}$ is calculated to be 6.0. The quantization function $Q()$ is implemented using a look-up table for quantization in a custom CUDA kernel since the FP4 format supports only $2^4 = 16$ distinct values. Detailed format regulations and quantization implementation can be found in Appendix A.

## 3. Methodology

In a typical linear layer of a Transformer architecture, the computation can be expressed as $Y = A \cdot W$, where $A$ is the activation tensor and $W$ is the weight tensor. To fully leverage the capabilities of FP4 tensor cores, both $A$ and $W$ need to be quantized to FP4, as shown in Figure 2. However, directly quantizing these tensors into FP4 introduces significant quantization errors. To address this challenge, we propose the differentiable gradient estimator method for weight tensors (Section 3.1) and the outlier clamping and compensation method for activation tensors (Section 3.2) to mitigate these issues.

### 3.1. Differentiable Gradient Estimator

Quantization functions are inherently non-differentiable, preventing the reverse flow of the gradient during backpropagation. The widely used **S**traight-**T**hrough **E**stimator (**STE**) (Bengio et al., 2013) bypasses this issue by assuming that the gradient of the quantized tensor is equivalent to that of the original tensor. However, this simplification introduces inaccuracies in low-bit settings, as noted in prior studies (Yin et al., 2019; Gong et al., 2019).

To overcome these limitations, we propose a **D**ifferentiable **G**radient **E**stimator (**DGE**) that reduces estimation errors. **DGE** maintains direct quantization for forward computation to preserve hardware efficiency while introducing a gradient correction term derived from a differentiable approximation of the quantization function.

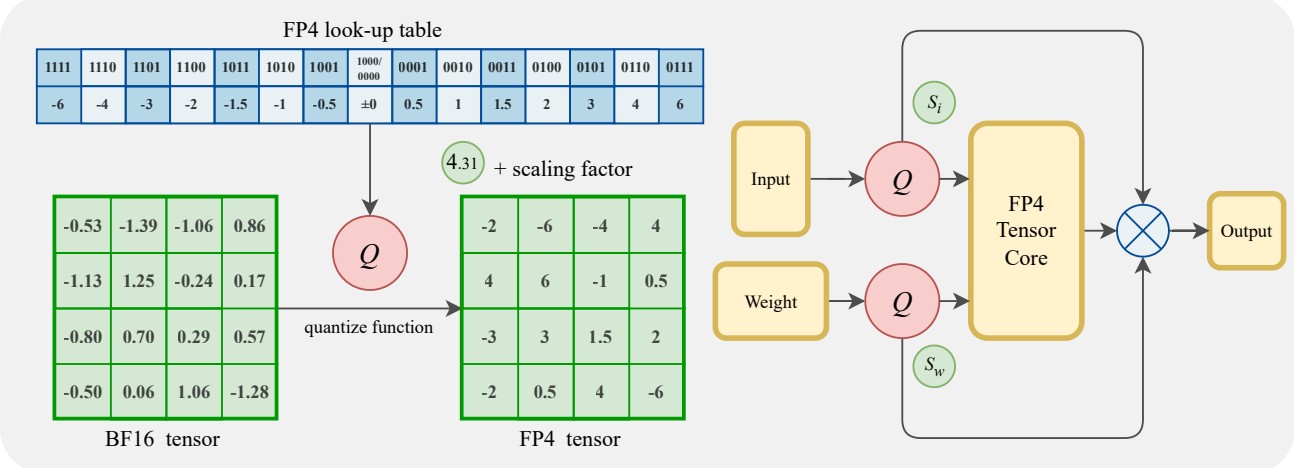

*Figure 2.* The structure of the proposed FP4 training scheme during the forward pass of a linear layer. A high-precision tensor, such as BF16, is quantized into the FP4 format using look-up table quantization. During the GeMM computation, both weight and activation tensors are quantized into FP4 to leverage the FP4 tensor cores. Two scaling factors are then applied to the final result to ensure computational correctness.

Suppose we quantize the model weight $W$ with a non-differentiable quantization function $f: W_q = f(W)$. Considering the backward gradient computation for a linear function with quantized weight, the forward pass can be expressed as:

$$Y = AW_q = Af(W) \tag{2}$$

During backpropagation, the loss gradient with respect to the weight $\partial L/\partial W$ and the activation $\partial L/\partial A$ are computed using the gradient propagated from the subsequent layer $\partial L/\partial Y$. For the weight gradient, the chain rule gives:

$$\frac{\partial L}{\partial W} = \frac{\partial L}{\partial W_q}\frac{\partial W_q}{\partial W} = (A^T \frac{\partial L}{\partial Y})\frac{\partial W_q}{\partial W} \tag{3}$$

Where $\partial W_q/\partial W$ represents the derivative of the quantization function $f$. since $f$ is an element-wise function, its derivative $f'$ is also element-wise. Thus we have:

$$\frac{\partial W_q[i,j]}{\partial W[k,l]} = \begin{cases} f'(W[i,j]), & \text{if } (i,j) = (k,l), \\ 0, & \text{otherwise.} \end{cases} \tag{4}$$

Therefore $\partial W_q/\partial W$ is a diagonal matrix. When applied to the chain rule Equation (3), this diagonal structure allows simplification of the gradient computation, reducing it to an element-wise multiplication between the two items:

$$\frac{\partial L}{\partial W}[i,j] = \frac{\partial L}{\partial W_q}[i,j] \cdot f'(W[i,j]) \tag{5}$$

or to be simplified:

$$\frac{\partial L}{\partial W} = \frac{\partial L}{\partial W_q} \odot f'(W), \tag{6}$$

Where $\odot$ denotes the element-wise (Hadamard) product.

Since $f$ is a non-differentiable quantization function, its derivative $f'$ is almost everywhere zero, leading to vanishing gradients and causing the weight gradient computation to fail, as shown in Equation (6). The **S**traight-**T**hrough **E**stimator (**STE**) addresses this issue by assuming $f'(W) \equiv 1$, thereby bypassing gradient vanishing. In other words, it directly assumes that $\partial L/\partial W \equiv \partial L/\partial W_q$.

To achieve more accurate gradient computation, we propose an alternative approach: approximating the quantization function with a well-chosen differentiable function, computing its derivative, and incorporating it into Equation (6). Specifically, we use the following function to simulate the quantization behavior:

$$f(x) = \frac{\delta}{2} \cdot \left(1 + \text{sign}(\frac{2x}{\delta} - 1) \cdot |\frac{2x}{\delta} - 1|^{\frac{1}{k}}\right) \tag{7}$$

Figure 3(a) illustrates this function under $k = 5$ for the range $[0, 0.5]$, which represents the first positive quantization interval in the E2M1 quantization scheme. This figure also shows that under the assumption of **STE**, forward quantization function is equivalent to $f(x) = x$ because $f'(x) \equiv 1$. In Equation (7), $\delta$ represents the quantization interval, and $k$ is a parameter that controls the degree of approximation. As $k$ increases, the function curve becomes sharper and more closely resembles the behavior of the original hard quantization function. For details on how Equation (7) is specified and derived, please refer to Appendix C.

The derivative of Equation (7) can be expressed as:

$$f'(x) = \frac{1}{k} \cdot |\frac{2x}{\delta} - 1|^{\frac{1}{k} - 1} \tag{8}$$

Figure 3(b) and Figure 3(c) show the complete quantization curve $f(x)$ and its derivative $f'(x)$ under $k = 5$ within

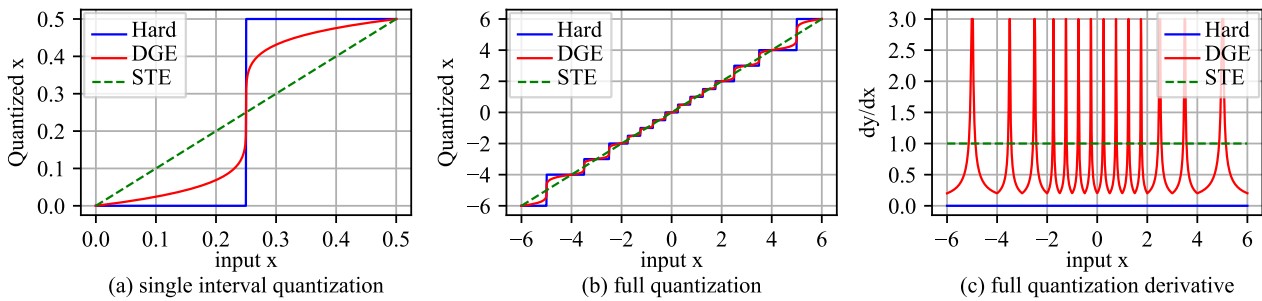

(a) single interval quantization     (b) full quantization     (c) full quantization derivative

*Figure 3.* Visualization of the **D**ifferentiable **G**radient **E**stimator (**DGE**). (a) Comparison of three quantization methods: hard quantization, differentiable quantization, and **STE** quantization, demonstrated on a single quantization step. (b) The full quantization curve for E2M1 quantization within its dynamic range $[-6.0, 6.0]$. (c) The derivative curves for the three methods, highlighting that hard quantization has a gradient of $f'(x) \equiv 0$ , while **STE** assumes a constant gradient of $f'(x) \equiv 1$.

the full E2M1 quantization framework. This framework consists of 14 distinct quantization intervals. In practice, the magnitude of $f'(x)$ is capped at 3.0 to prevent infinite gradient spikes at $\delta/2$ point, impacting only a very small subset of elements. For the mathematical soundness of this operation, as well as the supplementary integration process and proof for the **DGE** method in actual training process, please refer to Appendix C.

In practical model training, the **D**ifferentiable **G**radient **E**stimator (**DGE**) is seamlessly integrated into the process. During the forward pass, we retain the hard quantization function for computational efficiency. For the backward pass, a correction term derived from Equation (8) is applied to the weight gradient calculation following Equation (6).

### 3.2. Outlier Clamping and Compensation

During LLM training, activation tensors are significantly more challenging to quantize than weight tensors. This difficulty arises from the complex distribution of activation tensor values, often dominated by outliers—specific values that are substantially larger than the rest. Outliers pose a significant challenge to tensor quantization by disproportionately expanding the dynamic range of the target tensor, causing most values to underflow to zero after quantization.

To address this issue, we propose the **O**utlier **C**lamping and **C**ompensation method (**OCC**) to restrict the range of activation tensors and mitigate the underflow problem. Specifically, we identify outliers—values with the largest absolute magnitudes—through quantile identification and clamp them to a predefined threshold. Given a pre-defined quantile $\alpha$, the clamping function can be expressed as:

$$Y_c = \text{clamp}(Y, \max = \alpha, \min = 1 - \alpha) \qquad (9)$$

Figure 4 illustrates the impact of quantization with and

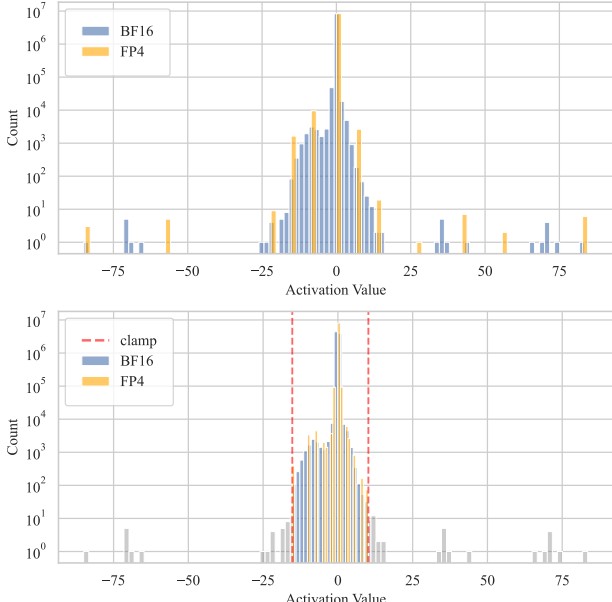

*Figure 4.* Visualization of the outlier clamping method, based on the first transformer layer's output of the LLaMA 1.3B model after 30,000 training iterations. Up: Quantization performed without outlier clamping, leading to severe loss of information. Down: Quantization after applying outlier clamping, effectively preserving tensor structure.

without outlier clamping, based on a real activation tensor extracted from the first transformer layer's output of the LLaMA 1.3B model after 30,000 training iterations, where $\alpha = 0.999$. This approach significantly reduces the mean squared error (MSE) between the original and quantized tensors, enhancing quantization quality and maintaining training stability.

We also observed that while clamping effectively reduces quantization error, it inherently introduces some error by

*Table 1.* Quantitative analysis of mathematical accuracy between original and quantized activation tensors. Results represent the average values obtained across all activation tensors on the 30,000 training iterations of the LLaMA 1.3B model.

| CLAMP | COMP | QUANTILE | SIM ↑ | MSE ↓ | SNR ↑ |
|-------|------|----------|-------|-------|-------|
| × | — | — | 92.19% | 0.1055 | 8.31 |
| √ | × | 99.9 | 98.83% | 0.0366 | 14.25 |
| √ | √ | 99.9 | 99.61% | 0.0245 | 15.31 |
| √ | √ | 99 | 100% | 0.0099 | 18.38 |
| √ | √ | 97 | 100% | 0.0068 | 20.88 |

disregarding the outlier values. To further preserve accuracy, we propose compensating for this error using a sparse outlier matrix. In our experiments, the quantile clamping threshold $\alpha$ is set relatively high (around $0.99 \sim 0.999$), making the residual matrix $\Delta Y = Y - Y_c$ highly sparse, with only about $0.2\% \sim 2\%$ non-zero elements. During computation, the clamped matrix $Y_c$ is processed using FP4 GeMM, while $\Delta Y$ is handled with high-precision sparse matrix multiplication.

Table 1 provides a quantitative analysis of cosine similarity (SIM), mean squared error (MSE), and signal-to-noise ratio (SNR) between the original activation tensors and quantized tensors. These results represent average values obtained across all activation tensors on the 30,000 training iterations of the LLaMA 1.3B model, demonstrating the impact of outlier clamping and compensation on preserving tensor fidelity during real model training. The data shows that outlier clamping significantly improves both cosine similarity and SNR. Moreover, incorporating outlier compensation further reduces quantization loss. Notably, lowering the quantile threshold increases the compensation scale, further reducing quantization loss. However, this introduces a trade-off between computational efficiency and numerical accuracy that must be carefully considered.

## 4. Experiment

In this section, we evaluate the proposed FP4 training framework across language models of various sizes. Section 4.1 details the implementation of our FP4 training framework, including the model architecture and hyperparameters. Section 4.2 presents the main results, showcasing training curves and zero-shot performance on downstream tasks. Finally, Section 4.3 provides ablation studies to further validate the effectiveness.

### 4.1. Experiment Setup

During LLM training, General Matrix Multiplication (GeMM) accounts for over 95% of the computational workload, with this proportion increasing for larger models. Consistent with prior works (Xi et al., 2023; Yang et al., 2020; Dettmers et al., 2022), we focus on 4-bit quantization for GeMM operations, a core feature of FP4 tensor cores. In a GeMM computation $Y = AW$, where $A$ (sequence length × input channels) is the activation tensor and $W$ (input channels × output channels) is the weight tensor, quantization is applied along distinct dimensions to align with matrix multiplication logic: $A$ is quantized token-wise (sequence length dimension), while $W$ is quantized channel-wise (output channels dimension). The aforementioned accuracy-preserving techniques are integrated to minimize quantization error. Since FP4 Tensor Cores are unavailable, we validate FP4 performance using Nvidia H-series GPUs' FP8 Tensor Cores, which encompass FP4's dynamic range and enable accurate simulation.

In mixed-precision training (Micikevicius et al., 2017), non-GeMM operations, which account for a minor computational fraction, are performed at higher precision to preserve accuracy. Following the framework in (Peng et al., 2023), we perform gradient communication in FP8 format to reduce bandwidth usage and adopt their mixed-precision Adam optimizer to conserve GPU memory. Gradients and first-order moments are stored in FP8, while second-order moments are stored in FP16. Remaining operations, comprising a smaller computational portion, are executed in FP16 or BF16 for stability and precision.

We adopt the widely recognized LLaMA 2 model (Touvron et al., 2023) as the primary model architecture. The training is conducted from scratch using the DCLM dataset (Li et al., 2024), a comprehensive dataset well-suited for language model pretraining. Hyperparameters remain consistent across precision settings for fair comparison. The learning rate follows a warm-up and cosine decay schedule, with the warm-up phase spanning 5% of total steps and the learning rate gradually decreasing to 10% of its peak over the remaining 90%. The peak learning rate is $3 \times 10^{-4}$, with a weight decay of 0.1. For the Adam optimizer, we use $\beta_1 = 0.9$, $\beta_2 = 0.95$, and $\epsilon = 1 \times 10^{-8}$. For special hyperparameters used in FP4 method, we use $k = 5$ for differentiable gradient estimator and select $\alpha = 0.99$ as the activation clamp and compensation quantile. Input sequences are fixed at 2048 tokens, and the batch size is 2048, comprising approximately 4M tokens.

### 4.2. Main Results

We validate the effectiveness of our proposed FP4 training framework by comparing it against the widely adopted BF16 mixed-precision training scheme. Figure 5 presents the training loss curves for LLaMA models (1.3B, 7B, and 13B) trained with BF16 and FP4 precision. All models are trained on 100B tokens using the same dataset and identical

*Table 2.* Zero-shot evaluation for downstream tasks between BF16 models and FP4 models under different model sizes.

| Model Size | Precision | **Average** | PiQA | Hellaswag | ObQA | Arc-C | Arc-E | BoolQ | LogiQA | SciQ | Lambada |
|---|---|---|---|---|---|---|---|---|---|---|---|
| 1.3B | BF16 | **53.23** | 71.11 | 50.80 | 36.60 | 36.69 | 68.60 | 57.83 | 30.26 | 83.30 | 43.84 |
| | FP4(Ours) | **53.13** | 70.89 | 50.82 | 36.20 | 36.86 | 67.47 | 58.23 | 29.49 | 83.90 | 44.30 |
| 7B | BF16 | **53.87** | 71.22 | 52.03 | 37.40 | 38.99 | 67.47 | 60.55 | 27.65 | 85.00 | 44.56 |
| | FP4(Ours) | **54.42** | 71.87 | 52.97 | 38.40 | 39.85 | 67.97 | 62.20 | 27.96 | 84.70 | 43.88 |
| 13B | BF16 | **54.44** | 72.80 | 53.56 | 38.60 | 38.82 | 67.97 | 57.40 | 29.65 | 86.30 | 44.87 |
| | FP4(Ours) | **54.95** | 73.78 | 54.12 | 39.60 | 39.68 | 67.89 | 55.90 | 30.88 | 85.80 | 46.89 |

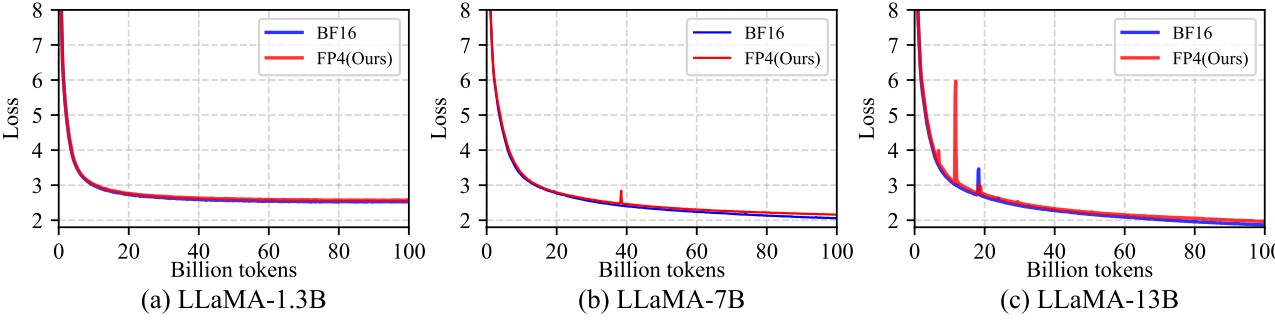

(a) LLaMA-1.3B      (b) LLaMA-7B      (c) LLaMA-13B

*Figure 5.* Training curves for BF16 models and FP4 models under different model sizes. (a) Training curves for 1.3B LLaMA model. (b) Training curves for 7B LLaMA model. (c) Training curves for 13B LLaMA model.

hyperparameters. The curves for BF16 and FP4 largely overlap across different model sizes, with the FP4 curve exhibiting a slightly higher training loss compared to the BF16 curve. Specifically, after training on 100B tokens, the training losses are as follows: 2.55 (FP4) vs. 2.49 (BF16) for the 1.3B model, 2.17 (FP4) vs. 2.07 (BF16) for the 7B model, and 1.97 (FP4) vs. 1.88 (BF16) for the 13B model.

In addition to training loss, we evaluate the models on a diverse set of downstream tasks datasets in a zero-shot manner, including Arc (Clark et al., 2018), BoolQ (Clark et al., 2019), HellaSwag (Zellers et al., 2019), LogiQA (Liu et al., 2021), PiQA (Bisk et al., 2020), SciQ (Welbl et al., 2017), OpenbookQA (ObQA) (Mihaylov et al., 2018), and Lambada (Paperno et al., 2016). These results are obtained through the widely used lm-evaluation-harness library[1] (Gao et al., 2024). As presented in Table 2, models pre-trained with FP4 demonstrate competitive performance in intrinsic in-context learning capabilities. Under the same model size, the average accuracy of FP4-trained models is comparable to, or even slightly exceeds, that of BF16-trained models. Additionally, the results follow the general trend: larger models achieve higher accuracy under the same number of training tokens.

---

[1] https://github.com/EleutherAI/lm-evaluation-harness

*Table 3.* Perplexity evaluation for downstream tasks between BF16 models and FP4 models under different model sizes.

| Size | Precision | **Average** | Lbd.OAI | Lbd.std | Pile10k | Wikitext |
|---|---|---|---|---|---|---|
| 1.3B | BF16 | **37.38** | 14.98 | 25.10 | 82.77 | 26.65 |
| | FP4(Ours) | **36.86** | 15.33 | 23.07 | 82.52 | 26.51 |
| 7B | BF16 | **35.06** | 14.34 | 23.33 | 77.72 | 24.86 |
| | FP4(Ours) | **35.62** | 14.29 | 24.42 | 78.42 | 25.36 |
| 13B | BF16 | **33.69** | 12.42 | 22.45 | 75.06 | 24.81 |
| | FP4(Ours) | **33.99** | 13.67 | 21.62 | 75.84 | 24.83 |

Table 3 further presents the perplexity (PPL) evaluation results for several downstream datasets including Lambada OpenAI (Lbd.OAI), Lambada standard (Lbd.std) (Paperno et al., 2016), the Pile 10k (Gao et al., 2020) and Wikitext (Merity et al., 2017). The results demonstrate that FP4 models achieve comparable or even slightly lower PPL than BF16 models. As expected, larger models achieve lower perplexity under the same training token budget.

These results highlight that despite the reduced precision, FP4 training achieves nearly equivalent performance to BF16 both in terms of training loss and downstream task accuracy, making it a promising approach for efficient training of large language models.

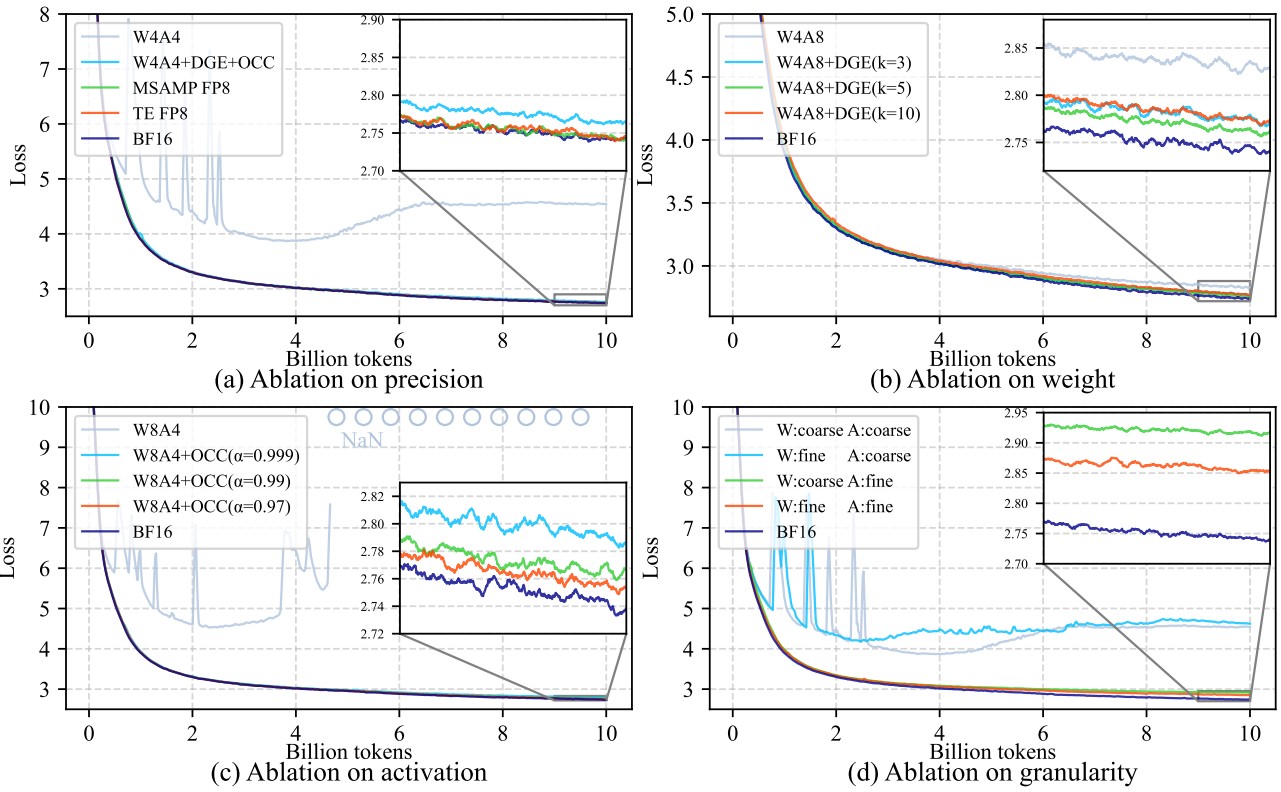

*Figure 6.* Ablation studies. (a) Training curves under different precision frameworks. (b) The effect of proposed **D**ifferentiable **G**radient **E**stimator (DGE). (c) The effect of proposed **O**utlier **C**lamping and **C**ompensation method (OCC). Note that directly casting activation into 4-bit leads to divergence, and the loss value turn into NaN (Not a Number). (d) Training curves under different quantization granularities of FP4.

### 4.3. Ablation Study

We divide our ablation study into smaller parts to better highlight the findings of FP4 training. All experiments are conducted on the LLaMA 1.3B model, trained with 10B tokens from a subset of the DCLM dataset. To accelerate convergence for this smaller model, the batch size is reduced from 2048 to 256, while other hyperparameters remain consistent with the main experiments.

**Precision**. Figure 6(a) presents training curves across various precisions, including BF16 (baseline), MS-AMP FP8 (Peng et al., 2023), Transformer-Engine FP8 (Nvidia, 2022), directly-casted FP4, and our FP4 method. We use W4A4 to denote direct quantization, meaning that quantizing both weight and activation to fp4. Meanwhile, W4A4+**DGE+OCC** denotes our fp4 quantization method that incorporates the **D**ifferentiable **G**radient **E**stimator (**DGE**) and **O**utlier **C**lamp and **C**ompensation (**OCC**) methods introduced in Section 3. The loss curves show that two FP8 methods and our FP4 approach maintain pretraining accuracy, while directly-casted FP4 has a significant training loss gap.

**Weights**. For weight-only 4-bit quantization (W4A8), we evaluate our **D**ifferentiable **G**radient **E**stimator (**DGE**) method alone against direct quantization. As shown in Figure 6(b), the **DGE** method significantly improve convergence. Notably, direct quantizing weight into 4-bit doesn't introduce a substantial training loss gap, suggesting that weights are easier to quantize than activations. For the hyperparameter $k$ in this method, a larger $k$ can better model the quantization function, but it can also lead to a more unstable correction term for the gradient. It can also be seen in the figure that a moderate $k = 5$ gives better final performance.

**Activation**. For activation-only 4-bit quantization (W8A4), we evaluate our **O**utlier **C**lamp and **C**ompensation (**OCC**) method alone against direct quantization. Figure 6(c) reveals that directly quantizing activations in FP4 results in curve divergence, where the loss values turn into NaN (Not a Number) after certain training steps. Outlier clamping and compensation effectively reduces this loss gap, ensuring a good convergence. This experiment re-emphasizes the importance of appropriate treatment of outliers in the absmax quantization framework. For the hyperparameter $\alpha$

in this method, a smaller $\alpha$ implies a stronger compensation, but at an increased computational cost. Figure 6(c) shows the model loss under three settings $\alpha = 0.999, 0.99, 0.97$, corresponding to the non-zero elements of the sparse compensation matrix of 0.2%, 2% and 6%, respectively. Although experiments show that a smaller $\alpha$ leads to better model accuracy, which is consistent with the conclusion of Table 1, we believe that $\alpha = 0.99$ is a better choice for comprehensive computational performance considerations.

**Granularity**. We also observe that the granularity of FP4 quantization plays a critical role. While FP8 training schemes (Peng et al., 2023; Nvidia, 2022) achieve sufficient accuracy with coarse-grained tensor-wise quantization, Figure 6(d) shows that tensor-wise scaling in FP4 introduces significant errors. To address this, we adopt vector-wise scaling, with token-wise quantization for activations and channel-wise quantization for weights, aligning with GeMM computation rules as discussed in Section 4.1. Notably, applying coarse-grained quantization to activations alone result in more severe accuracy degradation than applying it to weights alone, revealing that activations are harder to quantize than weights, consistent with the activation outlier issue described in Section 3.2.

## 5. Related Work

**Quantized Training and Inference**.When discussing the quantization of large language models (LLMs) for training, we typically refer to **Fully Quantized Training** (**FQT**). Related research efforts have generally used **Mixed Precision Training** (Micikevicius et al., 2017; Mellempudi et al., 2019) frameworks to accelerate model training while maintaining model accuracy. While previous research has mainly concentrated on CNNs or DNNs(Sun et al., 2019; Wang et al., 2018; Banner et al., 2018; Yang et al., 2020), recent studies have demonstrated the feasibility of low-bit mixed precision training for LLMs (Peng et al., 2023; Nvidia, 2022; Fishman et al., 2025; Xi et al., 2024). In contrast to the **FQT** scheme, research on low-bit computation for inference has focused on **Post-Training Quantization** (**PTQ**) and **Quantization Aware Training** (**QAT**). While **PTQ** directly quantizes pre-trained models for inference (Dettmers et al., 2022; Frantar et al., 2023; Lin et al., 2024a; Xiao et al., 2023; Yao et al., 2022; Liu et al., 2024), **QAT** involves fine-tuning or pre-training the model for better low-bit inference performance (Liu et al., 2023b; Cheng et al., 2023; Wang et al., 2023; Dettmers et al., 2023). Our method differs from **QAT**, as we aim to accelerate the training process while maintaining performance, rather than solely focusing on improving inference efficiency without consideration for the training speed.

**4-bit Quantization**. Recent works in **PTQ** and **QAT** have successfully applied 4-bit, 2-bit or even 1-bit quantization to

LLM inference (Dettmers & Zettlemoyer, 2023; Wu et al., 2023). **However**, these methods focused on LLM inference, requiring additional computation like calibration set fine-tuning (Wang et al., 2024), rotary matrix and low-rank compensation (Lin et al., 2024b; Ashkboos et al., 2024; Li et al., 2025), quantization parameters searching (Liu et al., 2023a), or even retraining the whole network (Ma et al., 2024). In the field of **FQT**, an early study (Sun et al., 2020) applied a 4-bit radix-4 FP4 format to convolutional neural networks (CNNs). MXFP (Rouhani et al., 2023b) introduced a novel quantization data for GPT-style models, but lacked feasibility validation on full FP4 settings. (Xi et al., 2023) proposed an INT4 training framework, but their focus was on fine-tuning tasks with limited applicability to LLM pretraining. In contrast, our work is the first to propose an FP4 training framework tailored for LLMs, validated from scratch, and designed to align with next-generation hardware like Nvidia's B-series GPUs.

**Differentiable Quantization**. Unlike previous methods focusing on differentiable quantization (Gong et al., 2019; Uhlich et al., 2019; Chen et al., 2019; Li et al., 2022; Huang et al., 2022), which rely on learnable quantization parameters updated through backpropagation, our differentiable gradient estimator method uses a fixed quantization function. We directly change the gradient estimator from **STE** to **DGE** during the backward pass, avoiding the need for continuous updates to the quantization function, which is not friendly to specialized hardware designs. Our approach is more efficient and more suitable for hardware acceleration in large-scale training.

**Handling Outliers**. Our method for handling activation outliers in LLMs differs significantly from existing approaches, which mainly target model inference (Liu et al., 2023a; Li et al., 2025; Ashkboos et al., 2024; Liu et al., 2024). Activation outliers in LLMs are typically channel-specific (Xiao et al., 2023; Wei et al., 2022). Channel-wise quantization would reduce quantization loss but conflicts with the computation structure of matrix multiplication in linear layers (Xi et al., 2024; Lee et al., 2024). Previous strategies to solve this problem like smoothing outliers (Xiao et al., 2023) or using rotary matrices (Ashkboos et al., 2024; Liu et al., 2024) rely on offline pre-processing, making them incompatible with pretraining tasks. In contrast, our method addresses outliers dynamically during real-time training without requiring separate calibration datasets, which is critical for maintaining efficiency in pretraining large models.

## 6. Limitation

One primary limitation of this work lies in the absence of dedicated FP4 Tensor Cores in existing hardware. Consequently, we are unable to directly measure the potential speedup and energy efficiency gains achievable with native

FP4 support. All current experiments rely on FP4 simulations, which introduce additional computational overhead due to extra precision casting and significantly prolong runtime. Additionally, due to constraints on computational resources, we have not yet extended our experiments to extremely large-scale models or to datasets comprising trillions of tokens. Investigating such scalability remain as critical directions for future research.

## 7. Conclusion

We propose the first FP4 pretraining framework for modern Large Language Models (LLMs), overcoming the challenges of limited dynamic range and quantization precision in 4-bit formats. By proposing a differentiable gradient estimator and an outlier compensation mechanism, we effectively reduce the accuracy gap between FP4 and higher-precision baselines like FP8 or FP16, achieving comparable performance across diverse model scales. Our findings demonstrate the feasibility of FP4-based training, providing insights into improving quantization methods for ultra-low-precision computing, and may also serve as a call for next-generation hardware designs to enable efficient 4-bit computation kernels.

## Impact Statement

This work demonstrates the feasibility of using ultra-low precision formats like FP4 for training large language models, offering a pathway toward energy conservation and reduced carbon emissions in AI development. By significantly lowering computational demand, FP4-based methods can democratize access to advanced AI systems while promoting environmental sustainability.

Additionally, this research calls for next-generation AI accelerators optimized for 4-bit computations, potentially shaping future hardware innovations. However, broader societal implications must be considered, including the risks of misuse and the amplification of biases inherent in large-scale AI models. Addressing these challenges is essential to ensure responsible and equitable adoption of this technology.

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

## A. Implementaiton of FP4 Quantizaiton

Floating-point numbers in a computer are represented using a binary format defined by the IEEE 754 standard (Kahan, 1996). Each number is divided into three components: the sign bit ($S$), the exponent ($E$), and the mantissa (or significand, M). This is commonly represented as ExMy, where x and y denote the number of bits for the exponent and mantissa, respectively. The sign bit determines whether the number is positive ($S = 0$) or negative ($S = 1$). The exponent, stored in a biased format, encodes the power of two that scales the number, enabling the representation of a wide range of values. The mantissa contains the significant digits of the number, capturing its precision. A normalized floating-point number is decoded as:

$$\text{Value} = (-1)^S \times (1.M) \times 2^{E-\text{bias}}$$

Where $1.M$ represents the normalized mantissa with an implicit leading 1, and the bias (e.g., 127 for single precision or 1023 for double precision) adjusts the exponent to account for its encoding. Subnormal numbers, where the exponent is all zeros, are handled separately with no implicit leading 1. This representation allows for efficient computation but introduces rounding errors due to the limited number of bits in the mantissa.

The IEEE 754 standard does not define rules for floating-point formats with precision below 16 bits, such as FP8 and FP4. For 4-bit floating-point representation, we adopt the E2M1 format as defined in prior studies (Rouhani et al., 2023b;a). According to the IEEE definition, an exponent field (E) filled with ones does not correspond to a valid numeric value; instead, it represents infinity (Inf) when the mantissa (M) is all zeros or an invalid number (NaN, Not a Number) when the mantissa contains nonzero bits. However, this rule is often disregarded in FP8 and FP4 formats due to their limited bit width, as the priority is to maximize the representation of meaningful numerical values. For example, FP8-E4M3 format doesn't define Inf, FP6 and FP4 formats don't define both Inf and NaN.

Based on the distribution of exponent and mantissa bits, all representable numbers in the FP4 format are listed in Table 4.

*Table 4.* FP4 Quantization Table under different FP4 formats.

| | BINARY SEQUENCE | | | | | | | | | | | | | | |
|---|---|---|---|---|---|---|---|---|---|---|---|---|---|---|---|
| **FORMAT** | 1111 | 1110 | 1101 | 1100 | 1011 | 1010 | 1101 | 1000/0000 | 0001 | 0010 | 0011 | 0100 | 0101 | 0110 | 0111 |
| E1M2 | -3.5 | -3 | -2.5 | -2 | -1.5 | -1 | -0.5 | ±0 | 0.5 | 1 | 1.5 | 2 | 2.5 | 3 | 3.5 |
| E2M1 | -6 | -4 | -3 | -2 | -1.5 | -1 | -0.5 | ±0 | 0.5 | 1 | 1.5 | 2 | 3 | 4 | 6 |
| E3M0 | -16 | -8 | -4 | -2 | -1 | -0.5 | -0.25 | ±0 | 0.25 | 0.5 | 1 | 2 | 4 | 8 | 16 |

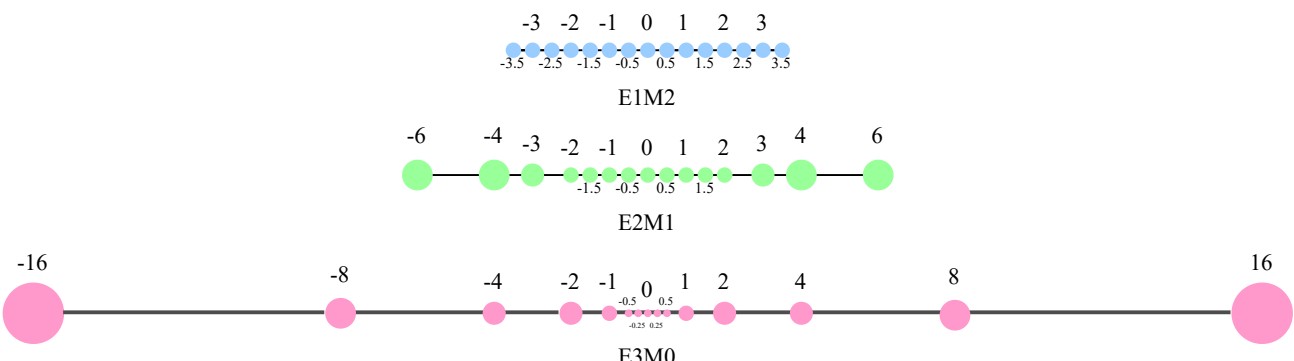

*Figure 7.* Visualization of all representable numbers in different FP4 formats.

The "E0M3" format is not included here because it is equivalent to the INT4 format, as it doesn't have any exponent bits. From Table 4 and Figure 7, we observe that increasing the number of exponent bits expands the dynamic range, while increasing the number of mantissa bits improves the precision of quantization intervals. We select the E2M1 format in our main experiments as it offers a balanced trade-off between dynamic range and quantization precision.

Since the FP4 format supports only $2^4 = 16$ distinct values, we implement a look-up table for FP4 quantization in a custom CUDA kernel. Quantization functions typically involve element-by-element operations on large amounts of data, which can

be parallelized to take advantage of the highly parallel computing power of GPUs. The following code paragraph shows the implementation of the quantization kernel.

```
1  __global__ void quantize_kernel(const float* x, float* output, int x_size) {
2      int idx = blockIdx.x * blockDim.x + threadIdx.x;
3      if (idx < x_size) {
4          float value = x[idx];
5          float closest;
6
7          closest = (value < -5.0f)  ? -6.0f :
8                    (value < -3.5f)  ? -4.0f :
9                    (value < -2.5f)  ? -3.0f :
10                   (value < -1.75f) ? -2.0f :
11                   (value < -1.25f) ? -1.5f :
12                   (value < -0.75f) ? -1.0f :
13                   (value < -0.25f) ? -0.5f :
14                   (value < 0.25f)  ? 0.0f :
15                   (value < 0.75f)  ? 0.5f :
16                   (value < 1.25f)  ? 1.0f :
17                   (value < 1.75f)  ? 1.5f :
18                   (value < 2.5f)   ? 2.0f :
19                   (value < 3.5f)   ? 3.0f :
20                   (value < 5.0f)   ? 4.0f : 6.0f;
21
22          output[idx] = closest;
23      }
24  }
25
26  void quantize(at::Tensor input, at::Tensor output, int size) {
27      const float* input_data = input.data_ptr<float>();
28      float* output_data = output.data_ptr<float>();
29
30      const int threadsPerBlock = 256;
31      const int blocks = (size + threadsPerBlock - 1) / threadsPerBlock;
32      cudaStream_t stream = at::cuda::getCurrentCUDAStream();
33
34      quantize_kernel<<<blocks, threadsPerBlock, 0, stream>>>(input_data, output_data, size)
           ;
35  }
```

## B. Theoretical Analysis on Speed Performance and Overhead

To assess the theoretical performance benefit of FP4 acceleration in our mixed-precision training framework, we analyze the theoretical speedup at the level of individual Transformer layer components. Specifically, we compute the floating-point operations (FLOPs) required for each subcomponent in both FP32 and FP4 precision. Let the hidden size be $h$, batch size $b$, and sequence length $s$. The FLOP breakdown per Transformer layer is summarized in Table 5.

Given that the backward pass typically incurs approximately twice the FLOPs of the forward pass, the total computational cost for both forward and backward passes is three times that of the forward pass alone. Thus, the theoretical speedup factor from FP4 acceleration (excluding any overhead) can be expressed as:

$$\frac{3 \times (24bsh^2 + 5bs^2h + 36bsh)}{3 \times (6bsh^2 + 5bs^2h + 36bsh)} = \frac{24h + 5s + 36}{6h + 5s + 36}$$

For a representative case of a 7B model with hidden size $h = 4096$ and sequence length $s = 2048$, the resulting theoretical speedup is approximately $\frac{24 \times 4096 + 5 \times 2048 + 36}{6 \times 4096 + 5 \times 2048 + 36} = 3.12$.

In addition to the theoretical speedup from FP4 compute, we also account for two primary sources of computational overhead introduced by our proposed methodologies: Differentiable Gradient Estimator (DGE) and Outlier Clamp and Compensation (OCC).

*Table 5.* FLOP breakdown of a standard Transformer layer under FP4 acceleration.

| Component | Subcomponent | FLOPs (FP32) | FLOPs (FP4) | Speedup Factor |
|---|---|---|---|---|
| Input LayerNorm | — | $4bsh$ | $4bsh$ | 1× |
| Multi-Head Attention | Query, Key, Value Projections | $6bsh^2$ | $1.5bsh^2$ | 4× |
| | Attention Scores Computation | $4bs^2h$ | $4bs^2h$ | 1× |
| | Softmax Computation | $bs^2h$ | $bs^2h$ | 1× |
| | Output Projection | $2bsh^2$ | $0.5bsh^2$ | 4× |
| Post-Attention LayerNorm | — | $4bsh$ | $4bsh$ | 1× |
| Feed-Forward Network (FFN) | Up Projection | $8bsh^2$ | $2bsh^2$ | 4× |
| | GeLU Activation | $28bsh$ | $28bsh$ | 1× |
| | Down Projection | $8bsh^2$ | $2bsh^2$ | 4× |
| **Total** | — | $24bsh^2 + 5bs^2h + 36bsh$ | $6bsh^2 + 5bs^2h + 36bsh$ | — |

**DGE Overhead.** DGE introduces an additional nonlinear function in the backward pass of GEMM operations used for weight updates. As shown in Equation (8), this adds approximately 8 FLOPs per input element. Accumulating over all relevant GEMM operations—including attention query/key/value projections, attention output projection, and MLP up and down projections—the total FLOP overhead per iteration is:

$$8 \times (3bsh + bsh + 4bsh + 4bsh) = 96bsh$$

This overhead occurs only once per forward-backward iteration.

**OCC Overhead.** OCC introduces extra FP8 sparse matrix multiplications during outlier compensation. Given an activation outlier matrix sparsity of $2(1 - \alpha)$, these sparse GEMMs are applied to each of the four primary GEMM computations in a Transformer block, resulting in an additional $2(1 - \alpha) \times (12bsh^2)$ FLOPs per iteration. In our setup, we choose $\alpha = 0.99$ to maintain high sparsity in the $\Delta Y$ matrix. While the computational FLOP overhead remains small due to high sparsity, hardware inefficiencies in sparse GEMMs make a high value of $\alpha$ essential for runtime performance.

**Adjusted Speedup Estimate.** Taking both DGE and OCC overheads into account, the revised theoretical speedup becomes:

$$\frac{3 \times (24bsh^2 + 5bs^2h + 36bsh)}{3 \times (6bsh^2 + 5bs^2h + 36bsh + 2(1-\alpha)(12bsh^2)) + 96bsh} = \frac{24h + 5s + 36}{6h + 24(1-\alpha)h + 5s + 68}$$

Substituting $h = 4096$, $s = 2048$, and $\alpha = 0.99$, we obtain $\dfrac{24 \times 4096 + 5 \times 2048 + 36}{6 \times 4096 + 24 \times 0.01 \times 4096 + 5 \times 2048 + 68} = 2.95$

**Overhead Impact.** The DGE overhead accounts for $32/(6h + 5s + 36) = 0.1\%$ and the OCC overhead contributes: $24(1 - \alpha)h/(6h + 5s + 36) = 5.6\%$ of the total computation. These result in a modest reduction of the ideal FP4 speedup from 3.12 to 2.95, which we consider an acceptable trade-off between computational efficiency and model accuracy.

## C. Supplementary Proof for Differentiable Quantization Estimator

### C.1. The Derivation of Differentiable Quantization Function

In Section 3.1, we introduce a differentiable function to simulate the quantization funciton in Equation (7). Here we describe in detail how this formula is derived step by step.

To construct a smooth and differentiable approximation to the quantization function, we begin with the power function $f(x) = x^a$, where $0 < a < 1$. This function is particularly suitable due to its saturating behavior: it approaches 1 as $x \to 1$, and rapidly decays toward 0 as $x \to 0$. These properties align well with the behavior of the right half of a typical

quantization function. To extend the function to the entire real line while maintaining central symmetry around the origin, we first symmetrize it by applying the sign function. This gives:

$$f(x) = \text{sign}(x) \cdot |x|^a \tag{10}$$

This symmetrized function now resembles a "soft step" centered at the origin, increasing smoothly from $-1$ to $+1$. We first scale the x-axis adjust the quantization interval from $[-1, 1]$ to $[-\delta/2, \delta/2]$, which we need to scale $x$ by $\delta/2$, resulting in:

$$f(x) = \text{sign}(\frac{2x}{\delta}) \cdot |\frac{2x}{\delta}|^a \tag{11}$$

To shift the center of this function from the origin to the midpoint of the quantization interval $[0, \delta]$, we perform a horizontal (x-axis) translation. Specifically, we translate the input by $\delta/2$, which centers the function at $x = \delta/2$. This gives:

$$f(x) = \text{sign}\left(\frac{2(x - \frac{\delta}{2})}{\delta}\right) \cdot \left|\frac{2(x - \frac{\delta}{2})}{\delta}\right|^a = \text{sign}\left(\frac{2x}{\delta} - 1\right) \cdot \left|\frac{2x}{\delta} - 1\right|^a \tag{12}$$

This modified function now transitions smoothly from negative to positive around the midpoint of the quantization interval, rather than around zero. Similarly, to ensure the output(y) range of the function maps to the desired quantization interval $[0, \delta]$, we apply a vertical (y-axis) translation and scaling. Specifically, we shift the function upward by 1 and then scale it by $\delta/2$, resulting in:

$$f(x) = \frac{\delta}{2} \cdot \left(1 + \text{sign}\left(\frac{2x}{\delta} - 1\right) \cdot \left|\frac{2x}{\delta} - 1\right|^a\right) \tag{13}$$

This transformation maps the function output to the interval $[0, \delta]$, with a steep but smooth transition occurring around $x = \delta/2$.

Finally, to make the steepness of the approximation more intuitively controllable, we reparameterize the exponent $a$ as the reciprocal of a positive constant $k > 1$, i.e., $a = 1/k$. Larger values of $k$ lead to steeper transitions, making the function better approximate a hard quantization step, while smaller values of $k$ result in smoother transitions.

### C.2. Proof of DGE with Vector-wise Scaling Factors

Here we present the complementary proof procedure for the **D**ifferentiable **G**radient **E**stimator (**DGE**) method under actual quantization with vector-wise scaling factors. In the GeMM operation $Y = AW$, where $A$ is the activation tensor with dimensions ($s \times c_i$, sequence length $\times$ input channels) and $W$ is the weight tensor with dimensions ($c_i \times c_o$, input channels $\times$ output channels), quantization is applied along distinct dimensions to adhere to the mathematical logic of matrix multiplication. For the weight tensor with dimensions ($c_i \times c_o$), channel-wise quantization is performed as follows:

$$W_{\text{scaled}} = W \odot sf \tag{14}$$

$$W_q^{\text{scaled}} = Q(W_{\text{scaled}}) \tag{15}$$

$$W_q = W_q^{\text{scaled}} \odot \frac{1}{sf} \tag{16}$$

Here, $sf$ is the scaling factor, and $\odot$ represents the element-wise (Hadamard) product. In tensor-wise quantization, $sf$ is a scalar. For channel-wise quantization, $sf$ is a vector with dimensions ($1 \times c_o$). In this case, the $\odot$ operation involves broadcasting $sf$ to each column of the matrix $W$ ($c_i \times c_o$), followed by element-wise multiplication.

For Equation (16), it is crucial to note that multiplying by $1/sf$ ensures mathematical correctness. Practically, however, this step is performed after the GeMM kernel execution. In other words, the quantized weight tensor provided to the GeMM

kernel is the scaled quantized weight tensor $W_q^{\text{scaled}}$ from Equation (15). Nevertheless, for mathematical analysis, the quantized weight tensor $W_q$ must be re-scaled.

In the backward computation, the loss gradient with respect to $W$ is derived from the forward operation $Y = AW_q$. According to the matrix multiplication rules for differentiation, the gradient $\partial L/\partial W_q$ is computed using the activation gradient $\partial L/\partial Y$ from the subsequent layer.

$$\textbf{Fwd: } Y = AW_q \quad \textbf{Bwd: } \frac{\partial L}{\partial W_q} = A^T \frac{\partial L}{\partial Y} \tag{17}$$

By applying the chain rule and referring to Equations (14) to (16), the relationship between $\partial L/\partial W_q$ and the actual weight gradient $\partial L/\partial W$ is established. According to Equation (16), the gradient $\partial L/\partial W_q^{\text{scaled}}$ can be expressed in terms of $\partial L/\partial W_q$ using the scaling factor $sf$:

$$\frac{\partial L}{\partial W_q^{\text{scaled}}} = \frac{\partial L}{\partial W_q} \odot \frac{1}{sf} \tag{18}$$

Subsequently, the differentiable gradient estimator correction term $Q'(x)$ is applied to compute $\partial L/\partial W_{\text{scaled}}$:

$$\frac{\partial L}{\partial W_{\text{scaled}}} = \frac{\partial L}{\partial W_q^{\text{scaled}}} \odot Q'(W_{\text{scaled}}) \tag{19}$$

Where $Q'(x)$ is the differentiable gradient estimator correction item introduced in Equation (8). Finally, the relationship between $\partial L/\partial W_{\text{scaled}}$ and $\partial L/\partial W$ is derived by incorporating $sf$:

$$\frac{\partial L}{\partial W} = \frac{\partial L}{\partial W_{\text{scaled}}} \odot sf \tag{20}$$

By combining all these steps, the formula for calculating the true weight gradient $\partial L/\partial W$ is obtained:

$$\frac{\partial L}{\partial W} = \left( \frac{\partial L}{\partial W_q} \odot \frac{1}{sf} \odot Q'(W_{\text{scaled}}) \right) \odot sf \tag{21}$$

$$= \frac{\partial L}{\partial W_q} \odot Q'(W_{\text{scaled}}) \tag{22}$$

Importantly, the scaling and un-scaling steps cancel each other due to the element-wise nature of the operations, resulting in a simplified expression. This final formula matches the previously demonstrated Equation (6) in the main body of the paper, with the only difference being that the variables within the **DGE** correction term must account for scaled weights. No changes are required for the $Q$ and $Q'$ functions.

### C.3. Mathematical Soundness of DGE Clipping

As introduced in Section 3.1, the differentiable quantization scheme includes a DGE correction term whose derivative can become unbounded near the midpoint of the quantization interval. Specifically, the derivative $f'(x)$ is clipped to a maximum value of 3.0 in practice to prevent instability during training. In this appendix, we provide a detailed mathematical justification for this clipping operation and explain how a smoothing technique can be used to ensure continuity and numerical stability.

The derivative of the DGE approximation is given by:

$$f'(x) = \frac{1}{k} \cdot \left| \frac{2x}{\delta} - 1 \right|^{\frac{1}{k} - 1} \tag{23}$$

as also stated in Equation (8). This expression arises from differentiating the power-based approximation of the quantization function centered at $x = \delta/2$, where $\delta$ is the quantization range, and $k > 1$ is a hyperparameter that controls the steepness of the transition.

The critical issue lies in the exponent $\frac{1}{k} - 1$, which is negative for all $k > 1$. Negative exponents in the form $|x|^\alpha$ with $\alpha < 0$ are known to diverge as $x \to 0$, due to the reciprocal relationship. For instance, when $k = 3$, the exponent becomes $-\frac{2}{3}$, leading to:

$$f'(x) = \frac{1}{3} \cdot \left| \frac{2x}{\delta} - 1 \right|^{-\frac{2}{3}} = \frac{1}{3 \cdot \left( \left| \frac{2x}{\delta} - 1 \right|^{2/3} \right)} \tag{24}$$

which becomes singular as $x \to \delta/2$. In this limit, the term $\left| \frac{2x}{\delta} - 1 \right| \to 0$, causing the denominator to vanish and $f'(x) \to \infty$. This unbounded growth can lead to gradient explosions during optimization, especially when many input values are near $\delta/2$.

To eliminate the singularity at $x = \delta/2$, we redefine the absolute value function in a differentiable and bounded manner. Instead of using the sharp absolute value $|x|$, we adopt a smooth approximation defined as:

$$|x| \approx \sqrt{x^2 + \epsilon^2} \tag{25}$$

where $\epsilon$ is a small positive constant. Substituting this into the derivative expression yields a smoothed version:

$$f'_{\text{smooth}}(x) = \frac{1}{k} \cdot \left( \sqrt{\left( \frac{2x}{\delta} - 1 \right)^2 + \epsilon^2} \right)^{\frac{1}{k} - 1} \tag{26}$$

This formulation ensures that the denominator never reaches zero, as $\sqrt{(\cdot)^2 + \epsilon^2} \geq \epsilon > 0$, and therefore $f'_{\text{smooth}}(x)$ remains bounded for all $x \in \mathbb{R}$. In effect, the singularity is replaced with a smooth transition that asymptotically approximates the original function as $\epsilon \to 0$, while maintaining differentiability and numerical stability during training.

We observe from the smoothed derivative expression presented in Equation (26) that the DGE correction term becomes bounded due to the regularization introduced by the epsilon term. Specifically:

$$\lim_{x \to \delta/2} f'_{\text{smooth}}(x) = \lim_{x \to \delta/2} \frac{1}{k} \cdot \left( \sqrt{\left( \frac{2x}{\delta} - 1 \right)^2 + \epsilon^2} \right)^{\frac{1}{k} - 1} \tag{27}$$

$$= \frac{1}{k} \cdot \left( \sqrt{0 + \epsilon^2} \right)^{\frac{1}{k} - 1} = \frac{1}{k} \cdot \epsilon^{\frac{1}{k} - 1} \tag{28}$$

In practice, the DGE correction item is clipped with a predefined value, which is proved in Equation (28) to be mathematically equivalent since $k$ and $\epsilon$ are all constents. This approach is aligned with the simplicity principle in algorithm design: it avoids the need to modify the functional form while still achieving equivalent mathematical behavior around the singular point. As demonstrated above, both the smoothed and clipped versions share the key property of bounding the gradient magnitude in the vicinity of $x = \delta/2$, thereby ensuring stability during backpropagation.

## D. Analyzing Quantization Difficulty Through Tensor Distribution

Section 3 highlights the necessity of quantizing both weight and activation tensors to fully leverage the FP4 tensor core. It also points out that activation tensors are significantly more challenging to quantize compared to weight tensors. To further support this observation, we provide the actual distributions of weight and activation tensors during model training.

Figures 8 to 10 illustrate the distribution of weight tensors, while Figures 11 to 13 show the distribution of activation tensors. These results are derived from training the LLaMA 1.3B model over 30,000 iterations. The y-axis is set to a logarithmic

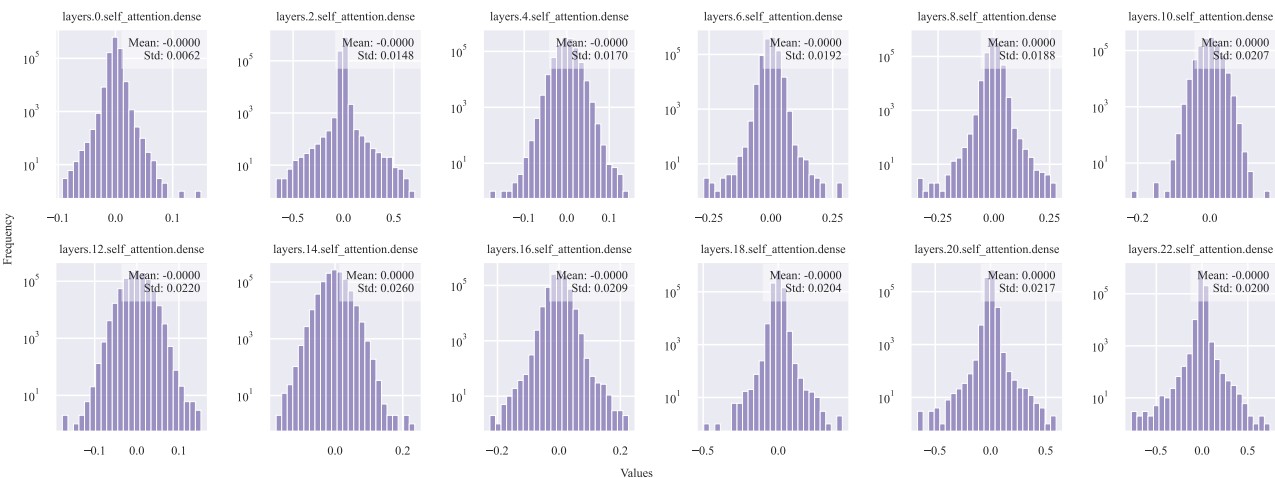

*Figure 8.* Visualization of the **weight** tensors in the dense projection layers of the self-attention module.

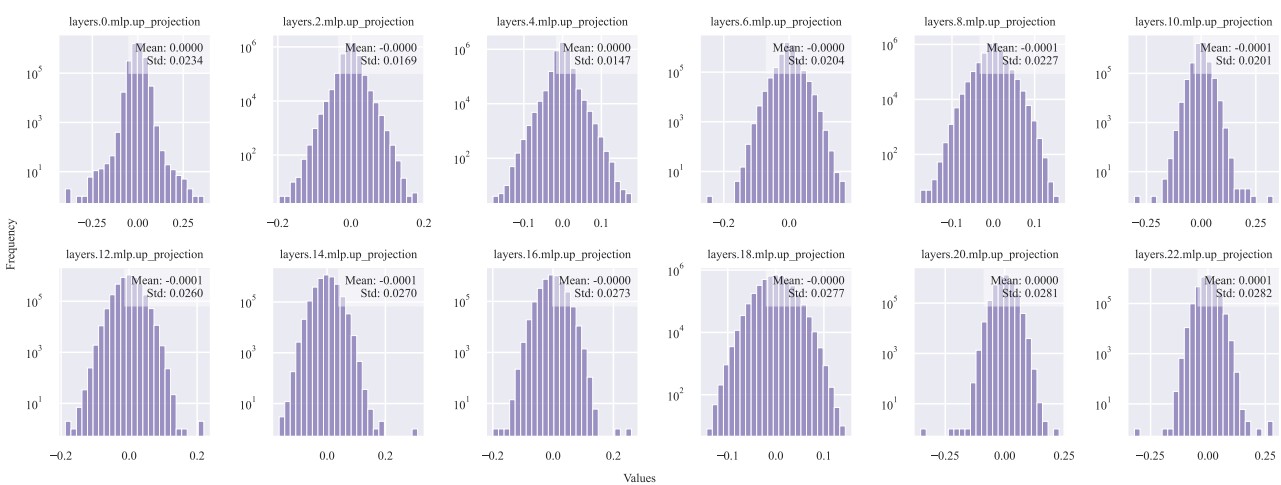

*Figure 9.* Visualization of the **weight** tensors in the up-projection linear layers of the MLP module.

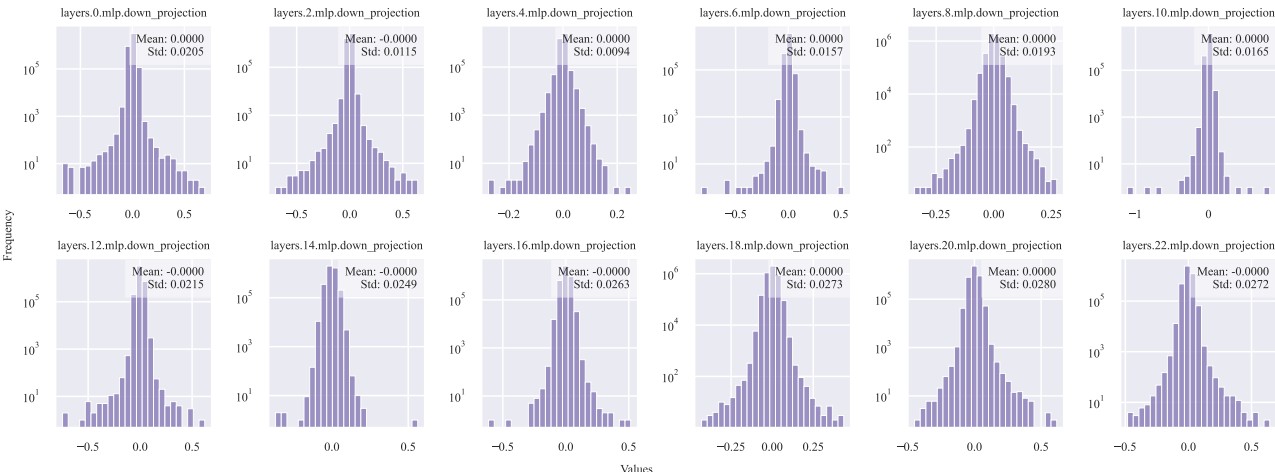

*Figure 10.* Visualization of the **weight** tensors in the down-projection linear layers of the MLP module.

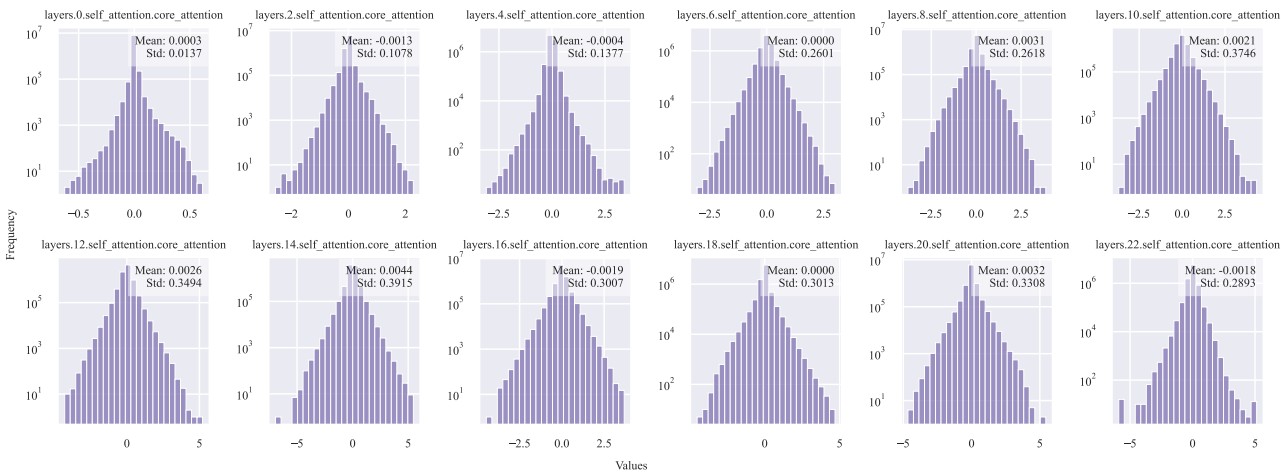

*Figure 11.* Visualization of the **activation** tensors from the core attention output.

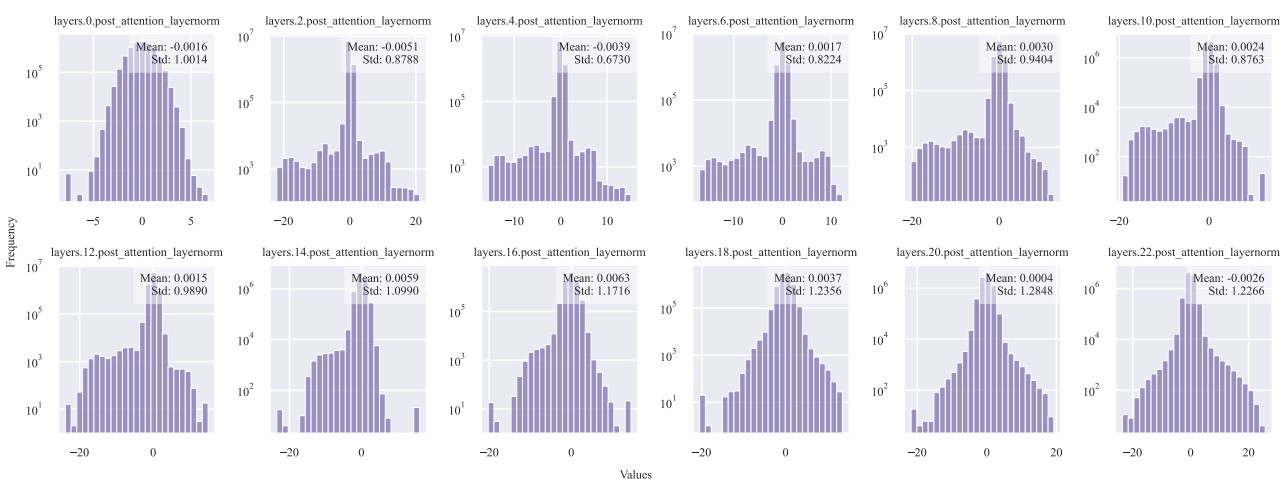

*Figure 12.* Visualization of the **activation** tensors from the post-attention layer normalization output.

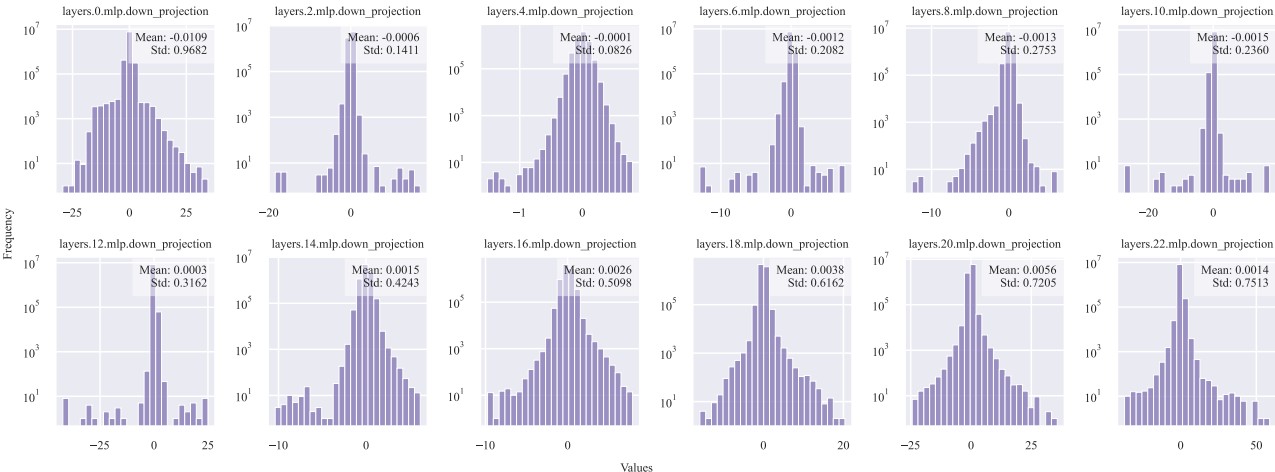

*Figure 13.* Visualization of the **activation** tensors from the MLP down-projection layer output.

scale to enhance visualization. From these figures, it is evident that weight tensors generally exhibit a smaller dynamic range, while activation tensors have a significantly larger dynamic range, making them more challenging to quantize.

Regarding distribution characteristics, weight tensors typically follow a normal distribution, with certain tensors exhibiting small outliers. In contrast, activation tensors vary widely in their distributions. For example, core attention outputs often follow a regular distribution with minimal outliers. However, many activation tensors, such as layer-norm outputs and transformer layer outputs, display irregular distributions with numerous outliers, making them particularly difficult to quantize.

Notably, the outliers in activation tensors during LLM training tend to appear in specific channels. This observation is further validated through heatmap analysis in Figure 14. The result is obtained through the activation function (GeLU) output from the first transformer layer.

These analyses underscore the critical importance of effectively addressing activation tensors during quantization, especially their outliers. Future research could gain valuable insights by exploring the complex distributions and outlier behavior of activation tensor values.

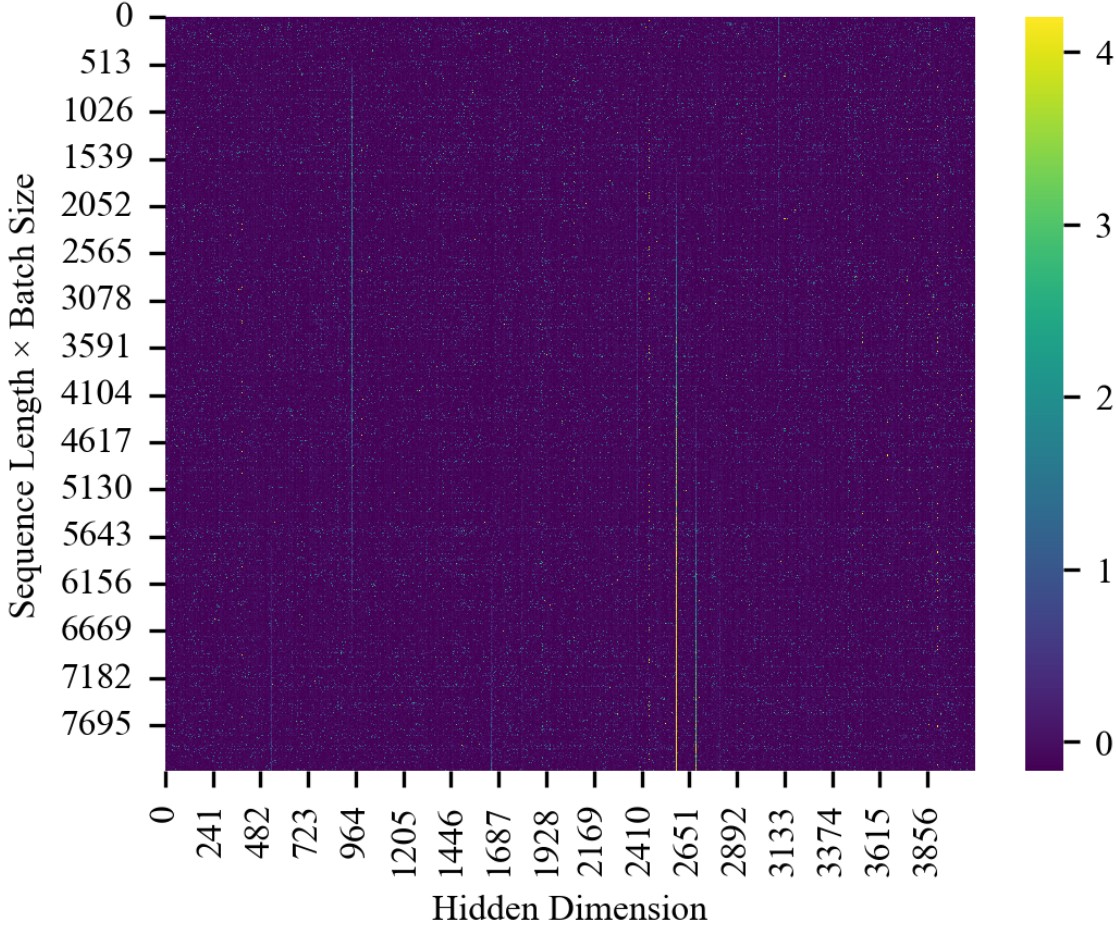

*Figure 14.* Heatmap visualization of the activation function (GeLU) output from the first transformer layer on the 30,000 training iteration of the LLaMA 1.3B model. The vertical light lines in the heatmap correspond to specific channel dimensions in the activation tensor, highlighting the channel-wise distribution of outliers.

