# OpenReview forum: "Optimizing Large Language Model Training Using FP4 Quantization"
_ICML.cc/2025/Conference — ICML 2025 poster_

### Official Review · Reviewer_gQsM · 2025-02-19

**Overall Recommendation:** 3

**Summary:**

This paper presents a FP4 training framework for large language models, which could potentially reduce the costs of LLM training. This work addresses the accuracy challenges caused by FP4 quantization with two key innovations: 1) Differentiable Gradient Estimator (DGE) for accurate weight updates and 2) Outlier Clamping and Compensation to tackle activation outliers. As a result, this framework maintains accuracy comparable to BF16 and FP8 while effectively scaling to models with up to 13B parameters.

**Claims And Evidence:**

Please refer to **“Methods And Evaluation Criteria”**.

**Essential References Not Discussed:**

N/A.

**Experimental Designs Or Analyses:**

Please refer to **“Methods And Evaluation Criteria”**.

**Methods And Evaluation Criteria:**

The accuracy evaluation and ablations appear to be plausible. It covers models with sizes from 1.3B to 13B and provides both training loss curves and downstream evaluations. However, to make the claim stronger, it would be better to have:
1) Perplexity (PPL) evaluation of the models trained with the proposed method and BF16 baseline, as PPL is more sensitive and commonly used than downstream evaluations.
2) Efficiency evaluation for this paper is lacking. Although FP4 tensor cores are not available, some analysis/estimations regarding the speedups would be helpful.

**Other Comments Or Suggestions:**

N/A.

**Other Strengths And Weaknesses:**

[Strength]
1. The paper targets an important topic: efficient training of LLMs. Despite room for further improvements, the evaluations and ablations appear to be plausible and comprehensive.
2. The DGE design in this paper seems interesting and effective for FP4 training.

[Weakness]
1. It would be better if the authors could estimate the speedups in the paper. While the FP4 tensor cores may not be available, the estimation of theoretical speedups can be obtained with kernel-level latency breakdown. For example, what are the latency proportions of FP8 GEMM kernels (for FP4 simulation) and quantization kernels in the training process?
2. Perplexity (PPL) evaluation of the models trained with the proposed method and BF16 baseline are not provided. As PPL is more sensitive and commonly used than downstream evaluations, it would be helpful to include PPL evaluation.

**Questions For Authors:**

1. According to Figure 2, the input activations and weights are quantized from BF16 to FP4 on-the-fly. As a result, the proposed method in this paper should not be able to reduce memory consumption during the training process. Is this contradictory with the claim in the Impact Statement that “By significantly lowering computational and memory demands, FP4-based methods can …” ?
2. The authors mention that activation is quantized token-wise and weight is quantized channel-wise. I would like to confirm that does it mean that each token in activation (each channel in the weight) has only one scaling factor? If so, is the per-channel/per-token max value calculation process included in the quantization kernel?

**Relation To Broader Scientific Literature:**

The paper follows the framework of [FP8-LM](https://arxiv.org/pdf/2310.18313), targeting the accuracy issues regarding FP4 LLM training.

**Theoretical Claims:**

I have checked the formulations for DGE in this paper. The overall derivation looks good to me. One question on Equation (8): Should there be a $\delta$ in the numerator of the constant term?

---

> ### Author Rebuttal · Authors · 2025-03-30
>
> We sincerely thank the reviewer for the thorough and constructive feedback on our work. We appreciate your recognition of the framework’s potential and your insightful questions, which have helped clarify key aspects of our methodology and claims. Below, we address each point in detail.
>
> **W1: Speedup Estimation and Kernel-Level Latency Breakdown**
>
> A1: We completely agree with the importance of evaluating theoretical speedup. Since we employ a mixed-precision training framework, we analyze the impact of FP4 acceleration on each computational component of a standard Transformer layer. Given a hidden size *h*, batch size *b*, and sequence length *s*, the FLOP breakdown is as follows:
>
> |Component|Subcomponent|FLOPs(FP32)|FLOPs(FP4)|Speedup Factor|
> |-|-|-|-|-|
> |Input Layernorm| |$4bsh$|$4bsh$|1x|
> |Multi-Head Attention|Query,Key,Value Projections|$6bsh^2$|$1.5bsh^2$|4x|
> | |Attention Scores Computation|$4bs^2h$|$4bs^2h$|1x|
> | |Softmax Computation|$bs^2h$|$bs^2h$|1x|
> | |Output Projection|$2bsh^2$|$0.5bsh^2$|4x|
> |Post-Attention Layernorm| |$4bsh$|$4bsh$|1x|
> |Feed-Forward Network(FFN)|Up Projection|$8bsh^2$|$2bsh^2$|4x|
> | |GeLU Activation|$28bsh$|$28bsh$|1x|
> | |Down Projection|$8bsh^2$|$2bsh^2$|4x|
> |Sum| |$24bsh^2+5bs^2h+36bsh$|$6bsh^2+5bs^2h+36bsh$| |
>
> Since the backward pass requires approximately twice the FLOPs of the forward pass, the theoretical FP4 speedup (excluding DGE and OCC overhead) for a 7B model (h=4096, s=2048) is:
>
> $$\frac{3*(24bsh^2+5bs^2h+36bsh)}{3*(6bsh^2+5bs^2h+36bsh)}=\frac{24h+5s+36}{6h+5s+36}=3.12$$
>
> The reviewer also point out the importance of kernel-level latency breakdown. Currently, we use FP8 GEMM kernels for all matrix multiplications in the above table, leading to a theoretical speedup of:
>
> $$\frac{3*(24bsh^2+5bs^2h+36bsh)}{3*(12bsh^2+5bs^2h+36bsh)}=\frac{24h+5s+36}{12h+5s+36}=1.83$$
>
> However, in practice, the framework of [FP8-LM](https://arxiv.org/pdf/2310.18313)  reports only a 1.28× actual speedup for h=4096 (7B model). Our implementation incurs additional precision casting overhead due to extra precision casting from BF16 to FP4 (for simulation) and FP4 to FP8 (for FP8 GEMM computation). These conversions are unnecessary with specialized FP4 hardware, where quantization could be fused into the GEMM kernel. So although we write the FP4 quantization kernel, it will still cause heavy overhead in the training process, reduceing speedup from 1.28× to approximately 1.08× in real training.
>
> **W2: Perplexity (PPL) Evaluation**
>
> A2: We agree that PPL is a sensitive metric for language model training. Below, we present the evaluation results:
> |Model|Precision|Lambada_openai|Lambada_standard|Pile10k|Wikitext|Average|
> |-|-|-|-|-|-|-|
> |1.3B|FP4|14.98|25.10|82.77|26.65|**37.38**|
> |1.3B|BF16|15.33|23.07|82.52|26.51|**36.86**|
> |7B|FP4|14.34|23.33|77.72|24.86|**35.06**|
> |7B|BF16|14.29|24.42|78.42|25.36|**35.62**|
> |13B|FP4|12.42|22.45|75.06|24.81|**33.69**|
> |13B|BF16|13.67|21.62|75.84|24.83|**33.99**|
>
> The results demonstrate that FP4 models achieve comparable or even slightly lower PPL than BF16 models. As expected, larger models achieve lower perplexity under the same training token budget.
>
> **Q3: For Equation (8), should there be a $\delta$ in the numerator of the constant term?**
>
> A3: Yes, it was a typo error and we're really sorry! We sincerely appreciate your attention to detail and will correct this in the final version.
>
> **Q4: Memory Consumption vs. Impact Statement Claim**
>
> A4: We appreciate the reviewer’s feedback on this potentially misleading claim. While our experiments confirm reduced GPU memory usage compared to BF16, this reduction primarily results from:
>
> 1. FP8 optimizer states in Mixed-precision optimizers.
> 2. FP8 activation storage (from FP8-LM framework).
>
> The current FP4 online quantization strategy does not reduce GPU memory. Further reductions would require FP4 optimizer states and activation storage, which we leave for future work. To prevent confusion, we will remove the memory claim from the Impact Statement in the final version.
>
> **Q5: Quantization Granularity**
>
> A5: Yes, each token (activation) or each channel (weight) has a single scaling factor. Currently the provided quantization kernel in Appendix A only covers the quantization of the scaled tensor, and per-channel/per-token max value computation is not yet fused into the kernel.
>
> We sincerely appreciate the reviewer’s insightful feedback, which has strengthened both our analysis and the clarity of our claims. We will incorporate all necessary revisions into the final manuscript.

---

### Official Review · Reviewer_1Hhi · 2025-03-12

**Overall Recommendation:** 4

**Summary:**

The paper tackles the challenging problem of FP4 training for LLMs. They propose two innovations: a differentiable quantization estimator (DGE) for precise weight gradient updates and an Outlier Clamping and Compensation (OCC) strategy for activations. The authors also provide extensive experiments on model scales (up to 13B parameters) show that the proposed FP4 method achieves comparable accuracy to traditional FP8 and BF16 baselines.

## Update after rebuttal
I maintain my original score. I am generally satisfied with the authors’ response.

**Claims And Evidence:**

The paper effectively supports its claims, showing through extensive experimentation that the FP4 quantization framework performs closely to higher precision baselines in training LLMs. The proposed differentiable gradient estimator (DGE) and outlier clamping and compensation (OCC) methods are convincingly validated by clear ablation studies (e.g., Figures 3, 4, and Table 1).

**Essential References Not Discussed:**

NA

**Experimental Designs Or Analyses:**

Experimental designs appear sound, but direct access to FP4 tensor cores is simulated due to hardware constraints. This limitation slightly impacts the empirical assessment of true hardware-level gains but cannot be attributed to the authors.

**Methods And Evaluation Criteria:**

The proposed evaluation criteria, such as training loss and zero-shot task performance, are appropriate for quantization research. The authors selected established benchmarks like PiQA, HellaSwag, ObQA, and Lambada, which are standard for assessing language models.

**Other Comments Or Suggestions:**

NA

**Other Strengths And Weaknesses:**

**Strengths**

- The proposed Differentiable Gradient Estimator (DGE) method is particularly innovative, providing a differentiable approximation that improves gradient accuracy significantly over the Straight-Through Estimator (STE).
- The Outlier Clamping and Compensation strategy is well-motivated and effective, especially given the difficulty of handling outliers in FP4 quantization.
- Clear presentation with illustrative figures and well-structured experimental analyses.

Authors have also addressed other limitations of their work in the paper such as using FP8 tensor cores to emulate FP4 computations due to hardware limitations

**Questions For Authors:**

1. Could you elaborate on the computational overhead introduced by the OCC method during actual training scenarios?

**Relation To Broader Scientific Literature:**

The problem is very relevant, especially with the recent interest in training LLMs in a resource-efficient manner.

**Theoretical Claims:**

There are no theoretical claims/proofs.

---

> ### Author Rebuttal · Authors · 2025-03-30
>
> We sincerely thank the reviewer for the thoughtful evaluation of our work and for recognizing the significance of our contributions. We appreciate your positive feedback on the effectiveness of DGE and OCC, as well as the thoroughness of our experiments. Additionally, we are grateful for your understanding of the hardware constraints that necessitated our simulation-based approach, which, while unavoidable, slightly limits empirical hardware-level assessments.
>
> **Q1: OCC’s Computational Overhead During Actual Training Scenarios**
>
> A1: The computational overhead of OCC primarily arises from additional sparse matrix multiplications. Specifically, the input activation tensor Y is decomposed as: $Y=Y_c+\Delta Y$, where $\Delta Y$ contains outlier values processed in higher-precision FP8 GeMM. Since $Y_c=\text{clamp}(Y, max=\alpha,min=1-\alpha)$ and $\alpha$ is very close to 1, the $\Delta Y$ matrix remains highly sparse, with a sparsity ratio of $2*(1-\alpha)$.
>
> A detailed theoretical analysis of the computational cost of $\Delta Y$ GeMM is provided in our response to reviewer iCFi (A2). We kindly refer you to those details if you're interested. The results show that under the training configuration of **hidden_size=4096, sequence length=2048 and $\alpha=0.99$** (as used inour 7B model training), the theoretical speedup of FP4 training decrease slightly from 3.12 to 2.95 due to OCC. Considering the large accuracy benefits of the OCC method, we think this performance drop is acceptable. Given OCC’s significant accuracy benefits, we consider this tradeoff acceptable. However, careful tuning of $\alpha$ is crucial to maintain high sparsity, as current hardware struggles with efficient sparse matrix multiplications at lower sparsity levels.
>
> However, the situation in real training scenarios is a little different. In our current training setup, FP8 GeMM is used to simulate FP4 behavior, and $\Delta Y$ is naturally handled in FP8 format. Consequently, both $Y_c$ and $\Delta Y$ are computed within FP8 GeMM, allowing their results to be directly summed without requiring an additional GeMM operation, thereby eliminating OCC-related overhead in this simulation setup.
>
> At present, the primary computational overhead stems from:
>
> 1. DGE computations (negligible).
> 2. Precision casting between BF16, FP4, and FP8 (significant).
>
> However, if native FP4 hardware becomes available and is used for $Y_c$ computations, then OCC’s computational cost must be carefully reassessed.
>
> Once again, we appreciate your constructive feedback and your recognition of this work’s broader relevance in advancing resource-efficient LLM training.

---

### Official Review · Reviewer_Le3f · 2025-03-12

**Overall Recommendation:** 3

**Summary:**

The paper introduces a framework for training LLMs using 4-bit floating-point (FP4) quantization to address the computational burdens of LLM training. It introduces a Differentiable Gradient Estimator (DGE) for weight updates and an Outlier Clamping and Compensation (OCC) strategy to manage activation outliers, mitigating quantization errors. The framework integrates mixed-precision training and vector-wise quantization. Experiments conducted on LLaMA2 models (up to 13B parameters,100B tokens) using FP8 tensor cores to emulate FP4 suggest performance comparable to BF16 and FP8 baselines in training loss and zero-shot tasks.

**Claims And Evidence:**

The core claim—that FP4 can train LLMs with minimal accuracy loss compared to BF16—is supported by training loss curves (e.g., 1.97 vs. 1.88 for 13B models) and zero-shot task accuracies (e.g., 54.95% vs. 54.44% for 13B). DGE and OCC are presented as solutions to FP4’s quantization challenges, with ablation studies showing their necessity (e.g., direct FP4 casting diverges). However, the evidence is undermined by the absence of native FP4 hardware testing, limiting efficiency claims to speculation.

**Essential References Not Discussed:**

The paper discusses a relatively comprehensive review of related work.

**Experimental Designs Or Analyses:**

Experiments test LLaMA 2 models (1.3B, 7B, 13B) on 100B tokens from the DCLM dataset, with ablation studies isolating DGE and OCC effects. While the design is systematic, its scale is insufficient—13B parameters and 100B tokens pale against modern LLMs (e.g., Llama 3’s15T tokens).

**Methods And Evaluation Criteria:**

The methodology employs mixed-precision training, quantizing GeMM operations to FP4 while using higher precision elsewhere. DGE approximates quantization differentiably, and OCC clamps outliers with sparse compensation. Evaluation includes training loss, quantization fidelity metrics (cosine similarity, MSE, SNR), and common zero-shot accuracy. Besides, the author need report the ppl metrics.

**Other Comments Or Suggestions:**

See above strengths and weaknesses.

**Other Strengths And Weaknesses:**

Strengths:

1. Interesting topic: This paper explored FP4 for LLM training, aligning with the push for future hardware trends (e.g., Blackwell GPUs).
2. Technical Effort: DGE are thoughtful attempts to address FP4’s quantization issues, supported by derivations (e.g., Equations 6-8) and ablations (Figure 6).
3. The comparison is extensive, including FP8-LM and TE, demonstrating the effectiveness of this paper.

Weaknesses:
1. Hardware Dependency: The lack of native FP4 hardware testing renders efficiency claims speculative, a drawback for a paper emphasizing training cost reduction.
2. Insufficient Scale: Experiments at 13B parameters and 100B tokens seems to outdate compared to current LLM scales (e.g., GPT-4’s 1T parameters/ more tokens).
3. Confused DGE: The direct presentation of Equation 7 is overly abrupt and somewhat confusing, lacking appropriate reasoning and motivation.
4. I understand that current hardware do not yet support FP4 tensor core; however, the authors should at least present the memory usage or training acceleration of existing methods. The absence of such data makes it difficult to accept their motivation for reducing training costs.
5. The novelty of OCC is straightforward and seems to lack novelty. Besides, author should present the computation burden of delta_Y.

**Questions For Authors:**

2. Does increasing the number of training tokens further still result in consistent convergence effects?
1.The implementation details will be useful. Could the authors consider open-sourcing the code for testing?

**Relation To Broader Scientific Literature:**

This work builds on FP8-LM , aiming to extend quantization to FP4 for training. This paper insufficiently engages with related FP4 works, like LLM-FP4.

**Theoretical Claims:**

This paper presents that FP4’s 16-value representation suffices for LLM training via DGE and managing outliers with OCC. However, The introduction of Equation 7 appears somewhat abrupt, and its specific definition, as well as the rationale for its particular form, can be quite confusing. These claims lack comparisons to other low-bit formats (e.g., INT4). The theoretical novelty is modest, leaning heavily on prior quantization concepts.

---

> ### Author Rebuttal · Authors · 2025-03-30
>
> We sincerely thank the reviewer for the thorough and constructive critique. We appreciate your recognition of the technical effort and alignment with hardware trends, as well as your insightful suggestions. Below, we address your concerns and questions:
>
> **W1: Hardware Dependency**
>
> A1: We acknowledge the lack of specific hardware as a limitation, as noted in Sec. 6. We provide a theoretical analysis of FP4 acceleration and demonstrate a **theoretical 3× speedup** (depending on model parameters). The detailed analysis is included in our response to reviewer gQsM (A1) due to space constraints. Additional analysis on DGE and OCC overhead, addressed in reviewer iCFi’s reply (A2), indicates the overhead remains acceptable while preserving accuracy. However, this is not speculative. As reviewer 1Hhi noted, while the absence of hardware testing affects empirical validation, it does not detract from our goal—to explore FP4 quantization for large-scale training in alignment with future hardware trends.
>
> **W2: Insufficient Scale (13B parameters, 100B tokens)**
>
> A2: We agree that scaling to trillion-parameter models is crucial. However, within academic compute constraints, our focus was to establish FP4’s feasibility. A **13B model trained on 100B tokens represents a reasonable scale in research**, even if it is smaller than industrial models (e.g., GPT-4, LLaMA3). Scaling stability is essential but is best explored in future industry-driven work.
>
> **W3: Clarification on DGE**
>
> A3: We apologize for not including the full mathematical derivation of Eq. 7 due to paper space constraints. We choose power function $f(x)=x^a$ (a<1) as the base differentiable estimation function because it saturates at 1 as x tends to 1, and quickly drops to 0 as x tends to 0, very much in line with the right half of the quantization function. To fully simulate the quantization function, we take absolute values of x and apply sign function for central symmetry: $f(x)=sign(x)\cdot|x|^a$. We then translate the function on the x-axis and the y-axis to move to the quantization range of [0, $\delta$]: $f(x)=\delta(1+sign(x-\frac{\delta}{2})\cdot|x-\frac{\delta}{2}|^a)$. Finally, we adjust the value of the power exponent to control the steepness and write a(a<1) as 1/k(k>1). We will clarify this in the final version to avoid possible confusion.
>
>
> **W4: Memory Usage and Training Speed**
>
> A4: We thank the reviewer for the suggestion. Below, we report real-time training memory usage and throughput:
> |Model Size|Precision|Memory Usage|Tokens per second|
> |-|-|-|-|
> |1.3B|BF16|51.03GB|470.3k (1.0x)|
> |1.3B|FP4|46.52GB **(-9%)**|395.1k (0.84x)|
> |7B|BF16|72.04GB|255.7k (1.0x)|
> |7B|FP4|52.75GB **(-27%)**|276.1k (1.08x)|
> |13B|BF16|70.28GB|118.4k (1.0x)|
> |13B|FP4|53.80GB **(-24%)**|126.9k (1.07x)|
>
> Note that our method is based on FP8-LM framework, where its speed up for three models are 1.03x, 1.24x and 1.27x (paper reported). In our implementation, **we need to do extra precision casting** for BF16 to FP4 (for simulation) and FP4 to FP8 (for FP8 GEMM computation), **leading to large overhead since native FP4 hardware is inaccessible**. In other words, evaluating speed performance during a real training process without dedicated hardware support can only serve as a point of reference.
>
> **W5: OCC’s Novelty and Computational Burden**
>
> A5: We thank the reviewer for pointing out that we did not state the OCC methodology very well in the paper. OCC’s novelty lies in its effective clamping strategies for characterizing numerical ranges and sparse compensation which avoids dense high-precision residuals. This method is suitable for online use and better matches the pre-training task. For the computation burden of $\Delta Y$, please refer to reviwer iCFi's reply (A2) due to the charactor limit. We'll refine these statements in the final version.
>
> **Q6: Perplexity (PPL) Metrics**
>
> A6: We acknowledge PPL’s importance as evaluation metric. Due to the character limitation, we kindly refer to reviwer gQsM's reply (A2) for detailed PPL results.
>
> **Q7: Convergence with Increasing Tokens**
>
> A7: Our preliminary analyses of existing scaling trends suggest that extended token training would likely achieve stable convergence. While scaling analysis is crucial for understanding model convergence, resource limitations currently prevent comprehensive token scalability studies. We highlight this as a critical research direction and commit to open-sourcing our framework to support collaborative exploration of token scaling dynamics.
>
> **Q8: Open-Sourcing the Code**
>
> A8: We sincerely appreciate the reviewer’s interest in implementation details. We are fully committed to open-sourcing the code and will release it once it has been thoroughly organized to ensure clarity and usability.
>
> Thank you again for your rigorous feedback. We hope these clarifications address your concerns.

---

> > ### Comment · Reviewer_Le3f · 2025-04-05
> >
> > Thank you for the author's response. Despite the hardware limitations and token scale, which prevented the authors from conducting actual tests on FP4, from my view, the topic and experimental results of this paper still make sense as an academic study. In particular, the authors should carefully revise the derivation process of DGE in the final version to make it clearer. Ultimately, I sincerely suggest that the authors open-source the code to enhance the reproducibility of this work.
> >
> > Overall, despite its shortcomings, the author's response has addressed my concerns. I have increased my score accordingly. Good Luck!

---

> > > ### Author Response · Authors · 2025-04-07
> > >
> > > Thank you for your constructive feedback and for recognizing the academic value of our work despite the hardware limitations. We sincerely appreciate your thoughtful suggestions, which will undoubtedly strengthen the final version of this paper. We will rigorously revise the derivation of DGE in the final manuscript. We also fully agree on the importance of reproducibility, and we will definitely release the implementation code publicly with detailed documentation.
> > >
> > > Thank you again for your valuable insights in refining this work, and also for your encouragement and support.

---

### Official Review · Reviewer_iCFi · 2025-03-18

**Overall Recommendation:** 4

**Summary:**

The paper presents an innovative framework for training large language models (LLMs) using FP4 quantization. The key contributions include:
1.Differentiable Quantization Estimator: This method improves gradient updates during FP4 computations by analyzing the impact of quantization on both forward and backward passes of neural networks, deriving a function with correction terms for accurate gradient estimation.
2.Outlier Clamping and Compensation Strategy: This addresses the issue of outlier values in activation tensors during LLM training, which can cause significant quantization errors. The strategy involves clamping these outliers to a predefined threshold and compensating for the introduced error using a sparse outlier matrix.
Detailed experiments are presented in the paper to validate the effectiveness of the framework:
1.Focused on 4-bit quantization for GeMM operations
2.Quantization is applied along distinct dimensions for activation and weight tensors, aligning with matrix multiplication logic.
3.The framework is tested on LLMs with up to 13B parameters trained on 100B tokens.
4.Results show that the FP4 framework achieves accuracy comparable to BF16 and FP8 with minimal degradation.

## update after rebuttal
Low-precision training has emerged as a clear trend in reducing computational costs for machine learning. After carefully considering the rebuttals from the authors and the feedback from other reviewers, I believe this work makes a significant contribution to both the academic community and the industry. Its technical insights and practical implications are particularly valuable in advancing the field of efficient training methodologies. Therefore, I will maintain my original rating of accept.

**Claims And Evidence:**

The claims are supported by clear and convincing evidence.

**Essential References Not Discussed:**

All important literatures are cited in this paper.

**Experimental Designs Or Analyses:**

The experimental designs and analyses in the paper appear sound and well-considered, including Comparison to Baselines/ Model Sizes and Training Data/ Training Loss Curves/ Downstream Task Evaluation/ Ablation Studies/ Quantization Granularity Analysis.
However, the authors do not give more details on speed evaluation. Though there is no hardware that support FP4 natively. A theoretical analysis on the computation overhead or other cost should be given.
Another issue is the spikes of the loss curve, Fig. 5. It seems more and larger spikes are observed for larger models. It's better to give the reasons, as it may indicate some potential risk scaling to larger models.

**Methods And Evaluation Criteria:**

The proposed methods and evaluation criteria are well-aligned with the problem of optimizing LLM training using FP4 quantization. They address the key challenges and provide a comprehensive assessment of the framework's effectiveness.

**Other Comments Or Suggestions:**

See above

**Other Strengths And Weaknesses:**

See above

**Questions For Authors:**

See above

**Relation To Broader Scientific Literature:**

The key contributions of the paper are closely related to the broader scientific literature in several specific ways:
1. Advancement of Quantized Training Methods
2. Differentiable Quantization Estimator
3. Outlier Clamping and Compensation Strategy
4. Mixed-Precision Training and Fine Grained Quantization
5. Hardware Considerations

**Theoretical Claims:**

I checked the derivation of DGE and OCC. Both of them are theoretically correct.
However, there is a small issue of OCC. In Sec. 3.2, the author provides a study of the influence of CLAMP/COMP/QUANTILE (Table 1). It can be seen that lower alpha value yields less loss, because that more values are moved to high precision residuals. However, in Sec. 4.3 (Activation part), they claim higher alpha leads to better model accuracy. This is inconsistent with that in Sec. 3.2. It should be clarified.

---

> ### Author Rebuttal · Authors · 2025-03-30
>
> We sincerely thank the reviewer for the thorough and insightful evaluation of our work, as well as for recognizing the theoretical and practical contributions of our FP4 training framework. Below, we address the questions and suggestions:
>
> **Q1 (Theroretical Claims): Clarification on OCC’s $\alpha$ value inconsistency (Sec. 3.2 vs. Sec. 4.3)**
>
> A1: Thank you very much for pointing out this issue. This is indeed a typo in Section 4.3—the correct statement should be that **a lower $\alpha$ leads to better model accuracy**. Theoretically, a lower $\alpha$ results in reduced quantization loss, as stated in Section 3.2 and shown in Table 1. Experimentally, Figure 6(c) in Section 4.3 confirms that lower $\alpha$ corresponds to a lower training loss curve. We will correct this in the final version. Again, we appreciate your keen observation!
>
> **Q2 (Experimental Designs Or Analyses): Details on speed evaluation / Theoretical analysis on the computation overhead or other cost**
>
> A2: Thank you for the suggestion. We recognize the importance of analyzing FP4’s theoretical speedup and computational overhead. The theoretical speedup of FP4 **(excluding DGE and OCC overhead)** is:
>
> $$\frac{3*(24bsh^2+5bs^2h+36bsh)}{3*(6bsh^2+5bs^2h+36bsh)}=\frac{24h+5s+36}{6h+5s+36}=3.12 \quad (h=4096, s=2048)$$
>
> For detailed analysis on the decomposition of computational components, we kindly refer the reviewer to the response to reviewer gQsM (A1) due to space constraints.
>
> **Regarding computational overhead:**
>
> - DGE overhead: DGE introduces an additional nonlinear function during GEMM backward for weight updates, adding ~8 FLOPs per input element (Eq. 8). Accumulating all GEMM operation, this will cause a total overhead of $8*(3bsh+bsh+4bsh+4bsh)=96bsh$ (Four additions in brackets are weight shapes of attn qkv proj, attn out proj, MLP up proj and MLP down proj). Note that this overhead occurs only once per forward-backward iteration.
>
> - OCC overhead: OCC incurs extra sparse matrix multiplication. FP8 sparse GEMM is used  with an activation sparsity of $2(1-\alpha)$. These FP8 sparse GEMM are added into every GEMM computation, adding an extra $2*(1-\alpha)(12bsh^2)$ FLOPs, where $12bsh^2$ comes from the cumulative sum of all GeMM calculations in the decomposition table that can be accelerated by FP4 (attn qkv proj, attn out proj, MLP up proj and MLP down proj), but in FP8 format with twice as many FLOPs as in FP4 $\bigg(2*(1.5bsh^2+0.5bsh^2+2bsh^2+2bsh^2)\bigg)$. Since we set $\alpha=0.99$, meaning these GEMMs are highly sparse, the overhead remains small, though hardware inefficiencies in sparse matrix multiplication necessitate choosing a larger $\alpha$ to ensure the high sparsity of the $\Delta Y$ matrix.
>
> Accounting for these overheads, the adjusted theoretical speedup is:
>
> $$\frac{3*(24bsh^2+5bs^2h+36bsh)}{3*(6bsh^2+5bs^2h+36bsh+2*(1-\alpha)(12bsh^2))+96bsh}=\frac{24h+5s+36}{6h+24*(1-\alpha)h+5s+68}=2.95 \quad (h=4096, s=2048, \alpha=0.99)$$
>
> The overhead from DGE and OCC acounts for $32/(6h+5s+36)=0.1\\%$ and $24(1-\alpha)h/(6h+5s+36)=5.6\\%$ of the total computation, respectively, reducing the theoretical speedup ratio for FP4 compared to BF16 reduced from 3.12 to 2.95. We believe this is an acceptable trade-off between accuracy and efficiency since they can largely reduce quantization errors during training.
>
> **Q3 (Experimental Designs Or Analyses): Loss curve spikes in Fig. 5 (larger models):**
>
> A3: Thank you for the insightful observation. Larger models (e.g., 13B) exhibit more pronounced loss spikes compared to smaller models (e.g., 7B, 1.3B). While similar spikes occur in BF16, they are more frequent and severe under FP4 due to its limited representation range and the larger number of elements per FP4 quantization vector, increasing quantization error. Addressing this may require more aggressive accuracy compensation strategies, like decreasing the parameter $\alpha$. Additionally, the issue may stem from the optimizer, as we currently use a FP8-based mix-precision optimizer. Larger models may demand higher precision for the optimizer.
>
> Thank you again for your valuable comments. We appreciate your constructive feedback, which has helped us clarify key aspects of our methodology and presentation.

---

### Decision · Program_Chairs · 2025-05-01

**Decision:**

Accept (poster)

**Comment:**

This paper proposes techniques to enable the training of large language models (LLMs) using FP4 weights and activations. The methods are applied to train LLMs with up to 13B parameters and on datasets of up to 100B tokens, demonstrating training dynamics comparable to BF16 baselines.

The paper is very clearly written and addresses an important topic. While the theoretical contributions and technical innovations are not particularly novel--the use of approximate gradients for step functions, quantile-based clamping, and high-precision sparse outlier matrices have been explored in prior quantization work--the strength of the paper lies in its experimental results. The framework demonstrates FP4 training with minimal degradation at a reasonably large scale, which is practically valuable and impactful. In addition, during rebuttal, the authors provide further clarification and additional analysis regarding computational overhead.

All things considered, the paper is recommended acceptance.